# The effects of 17α-estradiol treatment on endocrine system revealed by single-nucleus transcriptomic sequencing of hypothalamus

**Lei Li[1]\*[†], Guanghao Wu[2†], Xiaolei Xu[3†], Junling Yang[4], Lirong Yi[5], Ziqing Yang[6], Zheng Mo[3], Li Xing[3], Ying Shan[1]\*, Zhuo Yu[3]\*, Yinchuan Li[5]\***

[1]Department of Obstetrics and Gynecology, National Clinical Research Center for Obstetric and Gynecologic Diseases, Peking Union Medical College Hospital, Chinese Academy of Medical Sciences and Peking Union Medical College, Beijing, China; [2]School of Medical Technology , Beijing Institute of Technology, Beijing, China; [3]Department of Medical Oncology , Beijing Tsinghua Changgung Hospital, School of Clinical Medicine, Tsinghua University, Beijing, China; [4]Department of Cancer Biology and Pharmacology, University of Illinois College of Medicine, Peoria, United States; [5]Institute of Reproductive Medicine, Medical School of Nantong University, Nantong, China; [6]School of Basic Medical Sciences, Shandong University, Jinan, China

**\*For correspondence:**
lilei64@pumch.cn (LL);
shypumch@163.com (YS);
yza02214@btch.edu.cn (ZY);
18622397604@163.com (YL)

[†]These authors contributed equally to this work

**Competing interest:** The authors declare that no competing interests exist.

## eLife Assessment

This study demonstrates the potential role of 17α-estradiol in modulating neuronal gene expression in the aged hypothalamus of male rats, identifying key pathways and neuron subtypes affected by the drug. While the findings are **useful** and provide a foundation for future research, the strength of supporting evidence is **incomplete** due to the lack of female comparison, a young male control group, unclear link to 17α-estradiol lifespan extension in rats, and insufficient analysis of glial cells and cellular stress in CRH neurons.

**Abstract** This study investigated 17α-estradiol's effects on aged hypothalamic physiological activity via long-term administration. Single-nucleus transcriptomic sequencing (snRNA-seq) was performed on pooled hypothalami from each group: aged male Norway brown rats treated with 17α-estradiol (O.T), aged controls (O), and young controls (Y). Supervised clustering of neurons (based on neuropeptides/receptors) evaluated subtype responses to aging and 17α-estradiol. Aging-induced elevation of neuronal cellular metabolism, stress, and reduced synapse formation-related pathways were significantly attenuated by 17α-estradiol. Neuron population analysis showed that subtypes regulating food intake, reproduction, blood pressure, stress response, and electrolyte balance were sensitive to 17α-estradiol. 17α-estradiol increased serum oxytocin (Oxt) and hypothalamic-pituitary-gonadal (HPG) axis activity (elevated plasma Gnrh, total testosterone; reduced estradiol). Gnrh1 upregulation mediated its effects on energy homeostasis, neural synapse, and stress response. Notably, *Crh* neurons in O.T showed prominent stress phenotypes, distinct from *Agrp/Ghrl* neurons. Thus, HPG axis and energy metabolism may be key 17α-estradiol targets in male hypothalamus. Additionally, our results demonstrate that supervised clustering (based on neuropeptides/receptors) effectively assesses the responses of hypothalamic neuron subtypes to aging and 17α-estradiol treatment.

## Introduction

The hypothalamus serves as the central hub for controlling energy homeostasis, stress response, temperature, learning, feeding, sleep, social behavior, sexual behavior, hormone secretion, reproduction, osmoregulation, blood pressure, visceral activities, emotion, and circadian rhythms (*Hajdarovic et al., 2022*). The hypothalamic energy-sensing system, particularly the circuits that regulate food intake, plays a crucial role in lifespan extension (*Dacks et al., 2013*). Elevated metabolic activity in the aged hypothalamus has been reported in aged hypothalamus, including increased mTor signaling (*Masliukov, 2023*; *Yang et al., 2012*). Additionally, decreases in gonadotropin-releasing hormone (GnRH), Ghrh, Trh, monoamine neurotransmitters, and blood supply are hallmarks of aging hypothalamus (*Yang et al., 2023*).

Previous studies have demonstrated that 17α-estradiol extends the lifespan of male mice and has beneficial effects on metabolism and inflammation, similar to those of rapamycin and acarbose (*Stout et al., 2017*; *Shen et al., 2021*; *Wink et al., 2022*). Recent study indicated that 17α-estradiol also extends the lifespan of male rats (*Mann et al., 2020*). Further investigations revealed certain unique features of 17α-estradiol in life extension distinct to rapamycin and acarbose (*Watanabe et al., 2023*; *Burns et al., 2024*). Moreover, it has been shown that 17α-estradiol targets hypothalamic *POMC* neurons to reduce metabolism by decreasing feeding behavior through anorexigenic pathways (*Steyn et al., 2018*). Interestingly, the lifespan extension effect has only been observed in male animals (*Harrison et al., 2014*). The safety of 17α-estradiol is key for translation into clinical treatment, and the potential side effects on reproduction and feminization by 17α-estradiol treatment must be considered. However, contradictory results have been reported regarding its side effects on reproduction and feminization (*Stout et al., 2017*; *Isola et al., 2023*; *Stout et al., 2023*). Therefore, further investigation and verification are needed to understand the underlying mechanisms of lifespan extension and the safety of 17α-estradiol.

In this report, we utilized single-nucleus transcriptomic sequencing and performed supervised clustering of neurons based on neuropeptides, hormones, and their receptors. Supervised clustering offers better resolution in cell cluster screening compared to traditional unsupervised clustering. We assessed the effects of 17α-estradiol on metabolism, stress responses, ferroptosis, senescence, inflammation, and pathways involved in synaptic activity in each neuron subtype, ranking the most sensitive neurons. The effects of 17α-estradiol on reversing aging-related cellular processes were evaluated by two opposing regulatory networks involved in hypermetabolism, stress, inflammation, and synaptic activity. Several key endocrine factors from serum were examined, and the potential side effects of 17α-estradiol on specific neurons were also evaluated.

## Results

### The overall changes in aged hypothalamus with or without long-term 17α-estradiol treatment via snRNA-seq profiling

To investigate the hypothalamus as a potential key target of 17α-estradiol's effects on life extension, we performed snRNA-seq on the entire hypothalamus of aged and 17α-estradiol-treated aged Norway brown rats, using the hypothalamus from young adult male rats as a control. We identified 10 major cell types based on specific cell markers of the hypothalamus (*Figure 1A–B*). Notably, the proportions of all non-neural cells changed in O versus Y (*Figure 1C*). For instance, the proportions of oligodendrocytes (Oligo), oligodendrocyte precursor cells (OPC), and microglia (Micro) were found to be increased, while those of astrocytes (Astro), tanycytes (Tany), fibroblasts (Fibro), pars tuberalis cells (PTC), and endothelial cells (Endo) were decreased in O compared to Y. The proportions of Oligo, OPC, and Micro were also increased in 17α-estradiol-treated aged group (O.T) compared to those in Y. Furthermore, Endo was increased in O.T compared to both Y and O. The proportions of Astro, Tany, Epen, and PTC decreased more in O.T than those in O when compared to Y. These results indicated that 17α-estradiol treatment had extensive effects on the proportions of non-neural cells in hypothalamus.

Cell communication analysis revealed significant changes in the ligand-receptor pairs between neurons and other cell types, particularly those involving Endo, Fibro, Tany, and Astro (*Figure 1D*). Significant ligand–receptor interactions among neurons also changed in O.T and O groups, especially in O (*Figure 1E*). Notably, among the significant ligand–receptor pairs in neurons, *Bmp2–Acvr1/*

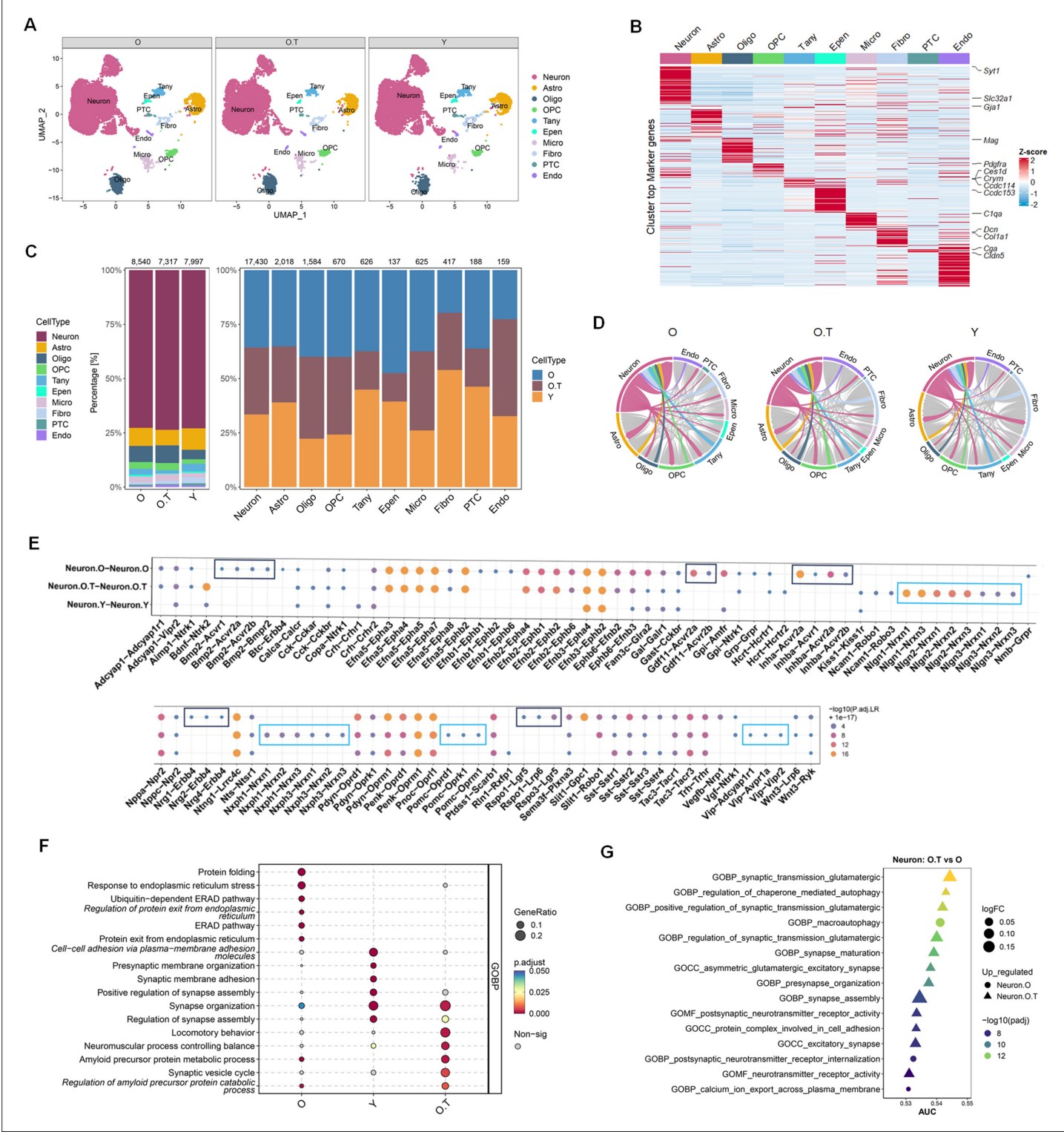

**Figure 1.** Single-nucleus transcriptomic sequencing (snRNA-seq) profiling of the hypothalamus from O, O.T, and Y samples. (**A**) UMAP visualization of nuclei colored by 10 cell types from hypothalamus of aged rats (**O**), 17α-estradiol-treated aged rats (**O.T**) and young rats (**Y**). (**B**) Heatmap showing the classic markers of 10 major cell types in hypothalamus. (**C**) Cell-type compositions by groups (left panel) or by major cell types with the total cell numbers shown above each column. (**D**) Circos plot depicting the number of ligand–receptor pairs between Neu and other cell types (color strips) for each group. (**E**) Dot plot showing significant ligand–receptor interactions between Neurons for each group. Boxes showing the unique ligand–receptor interactions between Neuron.O (black boxes) or between Neuron.O.T (blue boxes). (**F**) Dot plot of the top six enriched GO biological process terms

*Figure 1 continued on next page*

*Figure 1 continued*

across three groups of neurons via gene set enrichment analysis (GSEA) analysis. (**G**) The top 15 changed pathways/gene sets according to the ranks of AUC values in selected pathways related to neuronal synapses and axons from Gene Ontology (GO) biological process, GO molecular function and GO cellular component.

*Acvr2a/Acvr2b/Bmpr*, *Gdf11–Acvr2a/Acvr2b*, *Inhba–Acvr1/Acvr2a/Acvr2b*, *Nrg1/Nrg2/Nrg4–Erbb4*, *Rspo1–Lgr5/Lrp6*, and *Rspo3–Lgr5* were exclusively and significantly increased in neurons of the O group compared to those in O.T and Y, suggesting enhanced TGF superfamily-mediated signaling activity and canonical Wnt signaling during aging. The significantly changed ligand–receptor pairs *Nlgn1–Nrxn1/Nrxn2*, *Nlgn2–Nrxn1/Nrxn2/Nrxn3*, *Nlgn3–Nrxn1/Nrxn2/Nrxn3*, *Nxph1–Nrxn1/Nrxn2/ Nrxn3*, *Nxph3–Nrxn1/Nrxn2/Nrxn3*, *Pomc–Oprd1/Oprk1/Oprm1*, and *Vip–Adcyap1r1/Avpr1a/Vipr2* were exclusively increased in neurons of O.T compared to O and Y (*Figure 1E*). These ligand–receptor pairs were associated with synaptic activity, cellular adhesion, the opioid system, and vasodilation, indicating unique roles of 17α-estradiol in restoring certain physiological functions in the aging hypothalamus. The increased *Pomc* signal in O.T neurons aligns with previous reports that 17α-estradiol treatment decreases food intake in mice; this is likely because *Pomc* neurons promote satiety, and elevated *Pomc* signaling in O.T may enhance satiety-driven reduction in food uptake (*Figure 1E*; *Steyn et al., 2018*).

Gene set enrichment analysis (GSEA) based on DEGs also corroborated the expression profiles related to stress responses and synapse-associated cellular processes in neurons across the three groups (*Figure 1F*). ROC analysis of significantly differently expressed pathways related to neural synapses, manually selected from Gene Ontology databases, indicated that most top-ranked pathways related to synapses, according to AUC values, were downregulated in aged neurons, while 17α-estradiol treatment reversed this trend (*Figure 1G*, *Supplementary file 1*).

Overall, these findings suggest that 17α-estradiol broadly reshapes cell populations, cellular communication, neuropeptide secretion, and synapse-related cellular processes in the aging hypothalamus, distinguishing it from both the young hypothalamus and the untreated aged hypothalamus.

## The two opposing signaling networks in regulating metabolism and synapse activity, which can be balanced effectively by 17α-estradiol

To monitor the metabolism and neural status affected by 17α-estradiol, we utilized the energy metabolism pathway MitoCarta OXPHOS subunits to calculate the positively or negatively correlated pathways in hypothalamic neurons (*Figure 2—figure supplement 1*). Our findings revealed that energy metabolism and synapse activity represent two opposing regulatory signaling networks in hypothalamic neurons, with 17α-estradiol strongly playing a significant role in balancing these networks (*Figure 2A*). At the core of these opposing signaling pathways are two categories of contrasting TFs (*Figure 2B*). For example, *Calr*, *Clu*, *Peg3*, *Prnp*, *Ndufa13*, *Actb*, *Ywhab*, *Nfe2l1*, *Mtdh*, *Npm1*, *Bex2*, *Aft4*, and *Maged1* were positively correlated with pathways involved in OXPHOS subunits, lysosome function, protein export, mTorc1 signaling, and the unfolded protein response (UPR) in O, O.T, and Y neurons, while showing negative correlations with pathways related to ubiquitin-mediated proteolysis, endocytosis, tight junctions, focal adhesion, axon guidance, and MAPK signaling. Additionally, TFs *Myt1l*, *Ctnnd2*, *Tenm4*, *Camta1*, *Med12l*, *Rere*, *Csrnp3*, *Erbb4*, *Jazf1*, *Dscam*, *Klf12*, and *Kdm4c* exhibited opposite correlation patterns with these selected pathways in O, O.T, and Y neurons. These TFs may take conserved roles in regulating the two opposing biological processes in hypothalamic neurons.

We then attempted to establish gene signatures to represent these two opposing signaling networks, thereby displaying the cell status of aging and evaluating the effects exerted by 17α-estradiol. To achieve this, we evenly divided the expression levels of each of the six selected pathways from the two opposing signaling networks into four quarters (c1-c4) among the mixed neurons from O, O.T, and Y, calculating the shared unique markers in each quarter (*Figure 2C and D*). From the distribution patterns, we observed that the proportion of neurons in O decreased from c1 to c4 in metabolic pathways (MitoCarta OXPHOS subunits and Hallmark mTorc1 signaling), while this trend was reversed in the opposing signaling pathways (GOBP synapse organization and KEGG MAPK signaling pathway) (*Figure 2C*). In contrast, in Y, this trend was opposite, suggesting the expression levels from the four quarters (c1-c4) of the two opposing signaling networks can be used to monitor

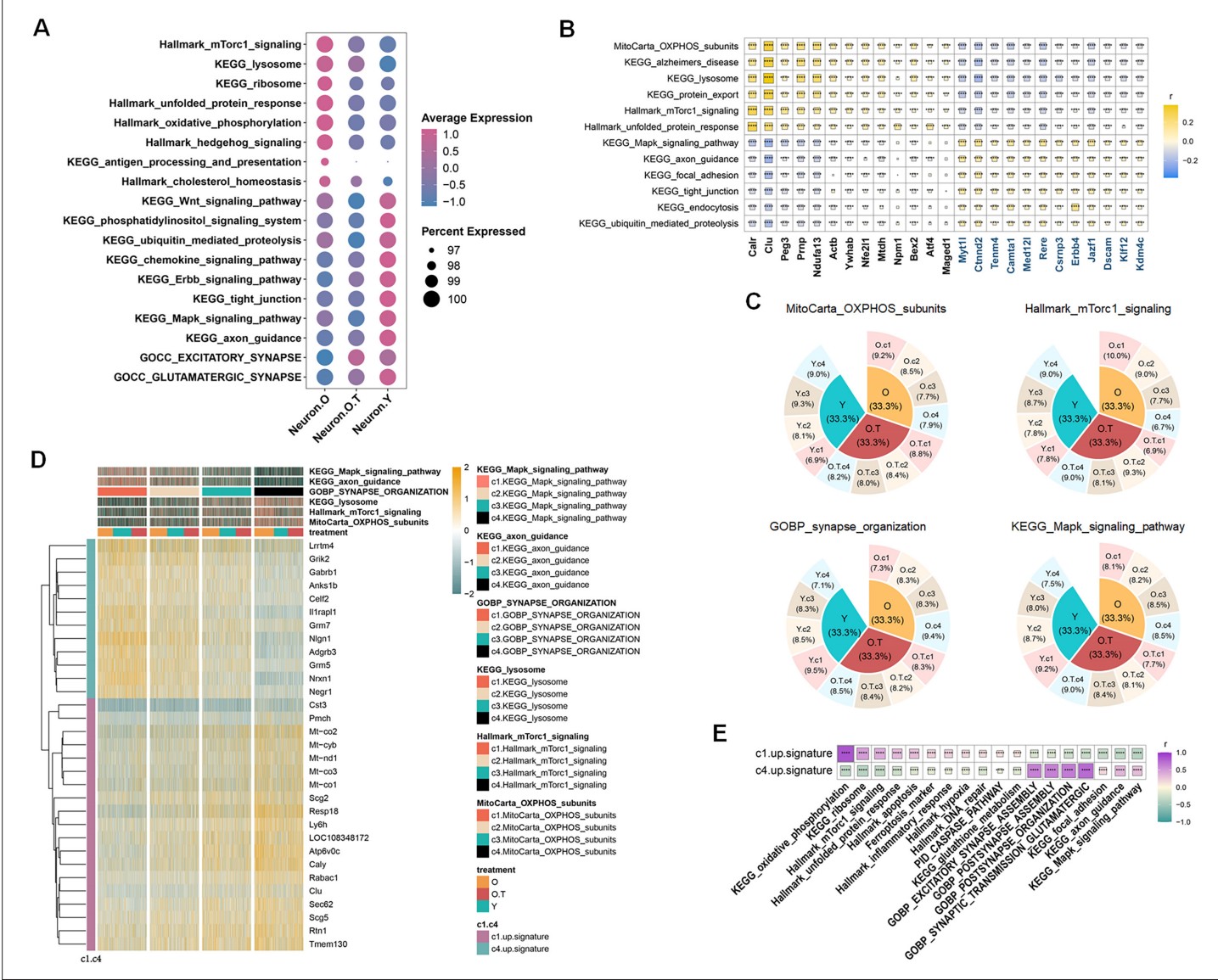

**Figure 2.** Two opposing regulatory signaling networks in neuron metabolism. (**A**) Dot plot of the selected pathways representing the prominent changes of overall expression levels across Neuron.O, Neuron.O.T and Neuron.Y in metabolism, signaling, and synaptic activity. (**B**) Correlation heatmap showing transcription factors (TFs) that correlated with the two opposing regulatory signaling networks in the mixed neurons of O, O.T, and Y. (**C**) The shared unique markers of each quarter (c1–c4) in six pathways in hypothalamic neurons (O, O.T, and Y). The markers were then collected as c1-up-signature (19 genes) and c4-up-signature (12 genes). (**D**) The aging-related cell proportions of each quarter are shown by four pathways. (**E**) The correlation of c1-up-signature and c2-up-signature with the two opposing regulatory signaling networks.

The online version of this article includes the following figure supplement(s) for figure 2:

**Figure supplement 1.** Top 20 signaling pathways or gene sets significantly positively or negatively associated with MitoCarta OXPHOS subunits in Neuron.O.

**Figure supplement 2.** The variable response patterns of non-neuron cells to aging and 17α-estradiol treatment in hypothalamus.

**Figure supplement 3.** The top enriched pathways of significantly expressed genes in Micro, Astro, and Neuron between O.T and O.

aging status. Treatment with 17α-estradiol alleviated this trend or even reversed it in O. We then screened the shared unique markers of each quarter from the six selected pathways in an attempt to establish the gene signatures representing the two opposing signaling networks. Unique markers in c1 (19 genes, c1-up-signature) and c4 (12 genes, c4-up-signature) were identified; however, c2 and c3 lacked unique markers shared by the six pathways (***Figure 2D***). Consequently, the 19 genes in c1-up-signature displayed an inverse correlation pattern with the 12 genes in c4-up-signature, indicating

the two opposing gene signatures are capable of reflecting the two opposing signaling networks in hypothalamic neurons (*Figure 2E*). Conversely, the balance of the two opposing signaling networks affected by 17α-estradiol in non-neural cell types was less pronounced than in neurons, showing variable effects on non-neural cells (*Figure 2—figure supplement 2*). GOBP pathway enrichment analysis revealed that Micro exhibited lower levels of synapse-related cellular processes in O.T compared to O, which was distinct from the observations in neurons (*Figure 2—figure supplement 3*). Therefore, in this report, we primarily focused on hypothalamic neurons and their responses to aging and 17α-estradiol.

## Supervised clustering revealed distinct responses of different subtypes of hypothalamic neurons to aging and 17α-estradiol

The hypothalamus contains numerous neuron subtypes that release various neuropeptides and hormones to regulate fundamental body functions. To differentiate the changes occurring during aging and the effect of 17α-estradiol on each neuron subtype, we performed supervised clustering based on neuropeptides, hormones, or their receptors (*Supplementary file 2*) (*Figure 3A*). The cell counts in each neuronal subcluster classified by neuropeptide secretion (neuropeptide-secreting subtypes) and subclusters defined by neuropeptide or hormone receptor expression (receptor-expressing subtypes) were quantified and compared in sample Y (*Figure 3B*). Notably, neurons expressing *Prlr*, *Esr1*, and *Ar* ranked among the top 20 receptor-expressing subtypes across all analyzed neuron populations. The similarity indices among these cell subtypes were further calculated (*Figure 3—figure supplement 1*), revealing high positive correlations in neuron subtypes expressing *Cartpt*, *Nxph4*, *Bdnf*, *Cck*, *Crh*, *Nppa*, *Adcyap1*, and *Penk*, as well as those expressing *Esr1*, *Calcrl*, and *Pth2r*. These similarities may partially reflect cellular overlap between subtypes (*Figure 3—figure supplement 1*).

We next calculated the prioritization of cellular perturbation induced by aging and/or 17α-estradiol treatment across these screened neuron subtypes (*Figure 3C and D*). The *Gnrh1* neuron subtype ranked among the top perturbed neuropeptide-secreting subtypes in both O vs Y and O.T vs Y comparisons (purple arrows). Notably, *Sct* and *Kiss1* neuron subtypes emerged as the top 2 perturbed populations in the O.T vs O analysis (red arrows), highlighting their heightened sensitivity to 17α-estradiol in the aged hypothalamus. Among receptor-expressing subtypes, *Insr* neurons showed the highest sensitivity to perturbation in both O vs Y and O.T vs Y comparisons (purple arrows, *Figure 3D*), while *Adipor2* and *Mlnr* neurons (blue arrows) ranked as the top 2 sensitive subtypes in the O.T vs O analysis. Intriguingly, neurons expressing *Ar* and *Esr1* ranked among the top 20 most perturbed receptor subtypes during aging (O vs Y), but were no longer ranked in this group following treatment (O.T vs Y and O.T vs O comparisons). This indicates that 17α-estradiol administration attenuated age-associated perturbation in these neuronal subtypes (*Figure 3D*).

## Differential senescence or stress levels and subtype-specific susceptibility in aged hypothalamic neurons

To gain a deeper understanding of the effects of 17α-estradiol treatment on the aged hypothalamus, we selected three gene signatures and two gene sets associated with aging, apoptosis, and stress to characterize the differential responses of distinct neuronal subtypes to aging and 17α-estradiol. These neuronal subtypes were then ranked separately based on neuropeptide-secreting subtypes and receptor-expressing subtypes (*Figure 4A and B*).

Neuropeptide-secreting subtypes, such as *Prlh-*, *Sct-*, *Gast-*, *Nppa-*, *Nxph1-*, *Ucn-*, *Pnoc-*, *Galp-*, and *Ghrl*-expressing neurons, were consistently ranked among the top 20 in at least 4 out of the 5 gene signatures or gene sets. These neurons are involved in gastrointestinal function, food intake, hunger, energy homeostasis, water homeostasis, vascular regulation, and pain, suggesting that aging exacerbates senescence or stress in these physiological processes.

In contrast, neurons expressing *Igf2*, *Crh*, *Npy*, *Npw*, *Npff*, *Nmu*, *Agrp*, or *Adipoq* ranked among the bottom 20 in at least four of the five signatures or gene sets. These neuropeptides and hormones are associated with cortical excitability, stress response, food intake, circadian rhythms, fat metabolism, insulin sensitivity, heart rate, and blood pressure. Notably, although *Crh*-expressing neurons exhibited high overall cellular perturbation among neuropeptide-secreting subtypes (*Figure 3C*), the relatively lower senescence and stress burden in *Crh-*, *Npy-*, *Npw-*, and *Nmu*-expressing neurons—key

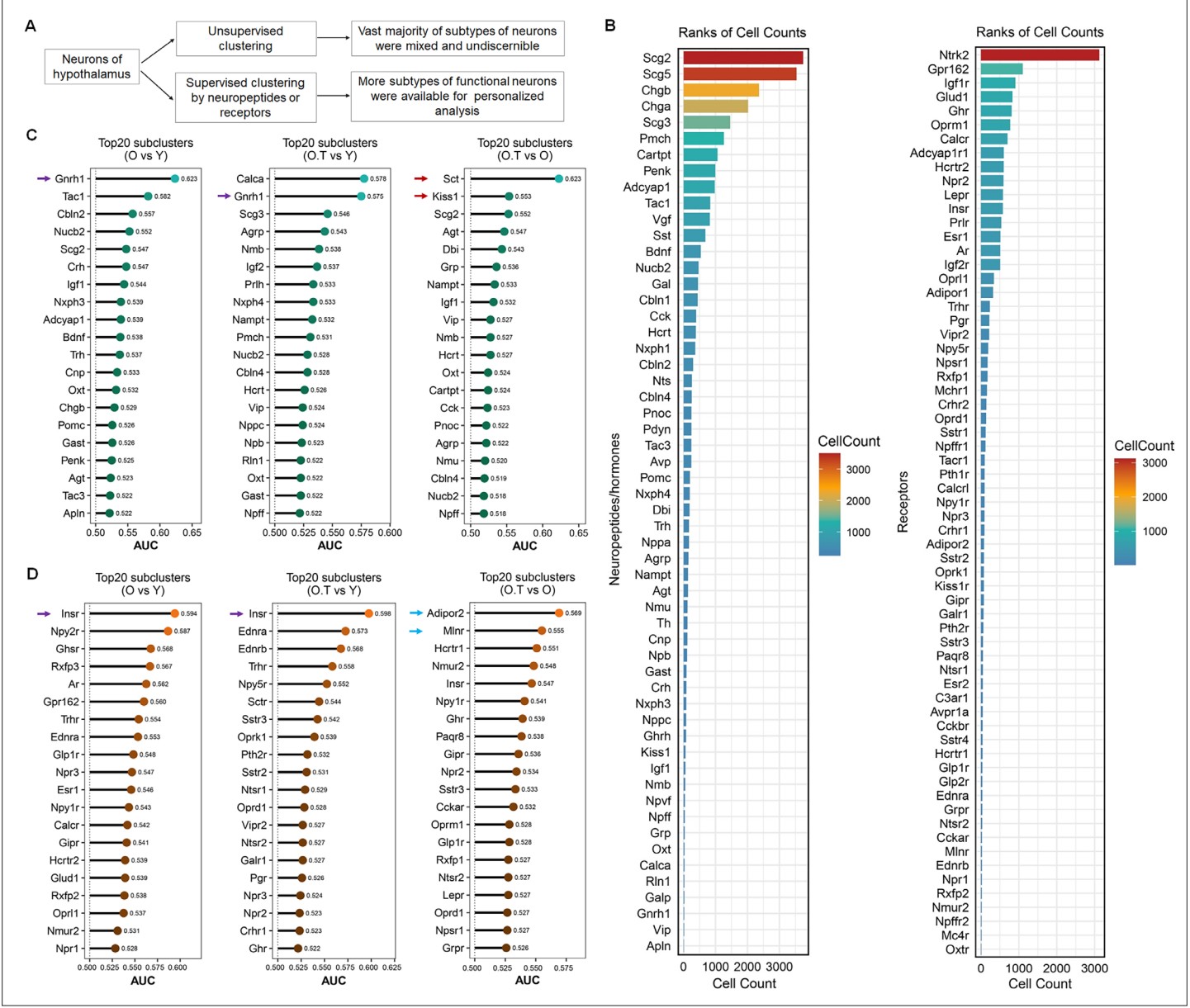

**Figure 3.** Screening of neuron subtypes via supervised clustering, which responded distinctly to aging and 17α-estradiol treatment. (**A**) Diagram outlining the features of supervised clustering of neurons in the hypothalamus in comparison with traditional unsupervised clustering. (**B**) The ranks of cell counts in neuropeptide-secreting neuron subclusters (left panel) and subclusters expressing neuropeptide receptors or hormone receptors (right panel) in sample Y. The cell number (n) in each subset is ≥10. (**C, D**) The prioritization of the top 20 neuron subclusters across the three types of perturbation (O vs Y, O.T vs Y, and O.T vs O) calculated by the Augur algorithm, in neuropeptide-secreting neurons (**C**) and neuron subclusters expressing neuropeptide receptors or hormone receptors (**D**).

The online version of this article includes the following figure supplement(s) for figure 3:

**Figure supplement 1.** The similarity of neuropeptide-expressing subclusters or receptor-expressing subclusters in young rat hypothalamus.

mediators of the stress response—compared to other neuronal subtypes represents a defining characteristic of the aged hypothalamus.

Regarding receptor-expressing subtypes, *Mc3r-*, *Sstr1-*, *Kiss1r-*, *Ntsr2-*, *Mlnr-*, *Ntsr1-*, *Npy1r-*, and *Avpr1a*-expressing neurons were consistently among the top 20 in at least 4 of the 5 gene signatures or gene sets. These receptor-expressing subtypes are involved in food intake, neurotransmission, reproduction, gut function, fat metabolism, circadian rhythm, and vasoconstriction, thereby indicating heightened stress in the aging hypothalamus. Conversely, *Glp2r-*, *Lepr-*, *Paqr8-*, and *Npr3*-expressing

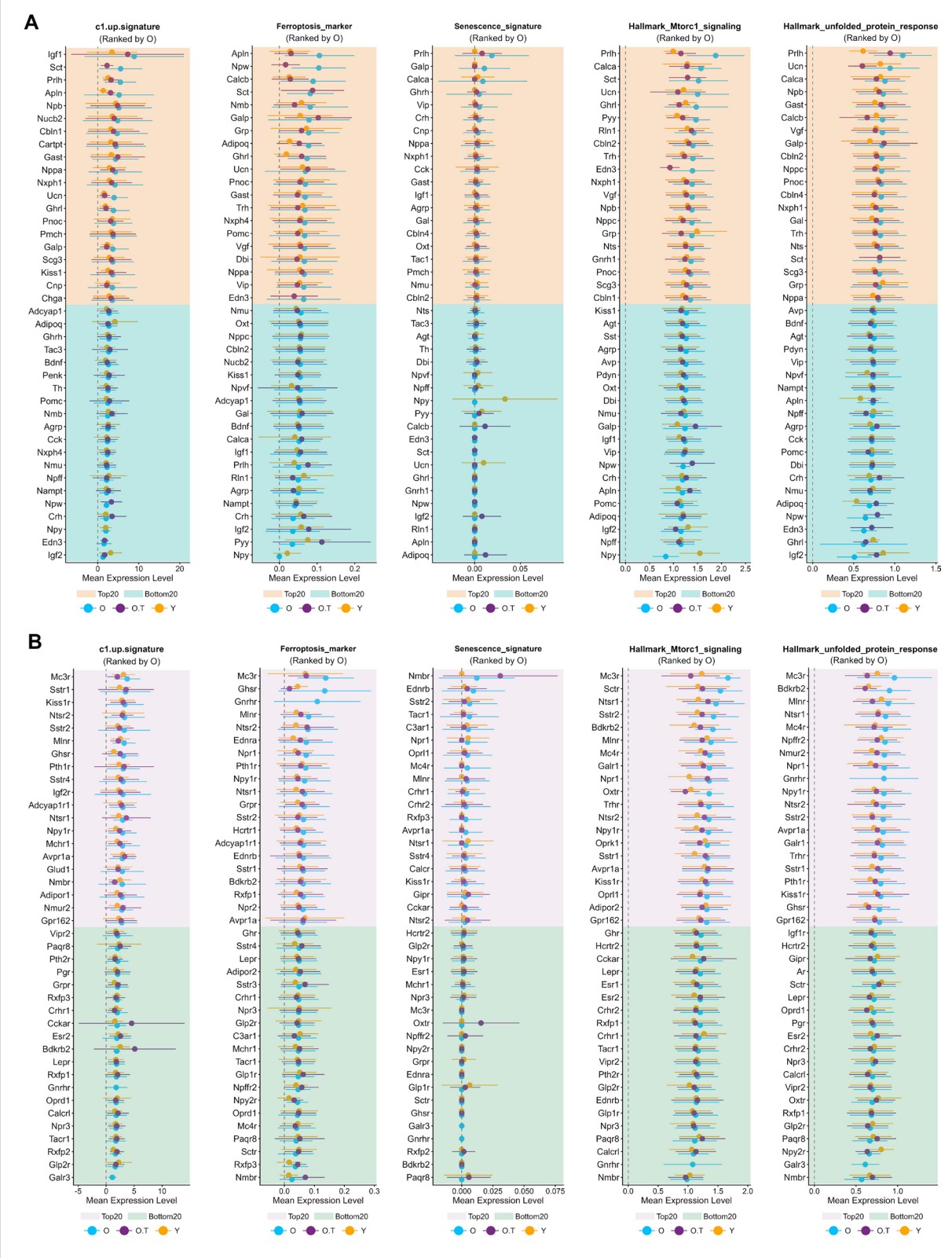

**Figure 4.** Ranking of neuron subtypes with distinct responses to aging and 17α-estradiol treatment. (**A, B**) The top 20 and bottom 20 neuron subtypes based on the mean expression values of five signatures or gene sets, ranked by their values in sample O, in neuropeptide-secreting subtypes (**A**) and in neuron subtypes expressing neuropeptide receptors or hormone receptors (**B**).

neurons were among the bottom 20 in at least 4 of the 5 gene signatures or gene sets, with associations to glucose regulation, fat metabolism, progesterone signaling, blood volume, and blood pressure.

Notably, most of the five signatures or gene sets in top-ranked neurons exhibited alleviated senescence or stress following 17α-estradiol treatment, indicating that such treatment mitigates senescence or stress in these specific neuronal populations (*Figure 4A and B*).

## The appetite-controlling neurons and hypothalamic–pituitary–adrenal (HPA) axis were altered by long-term 17α-estradiol treatment in the males

To further investigate the positive effects, potential side effects, or compensatory effects of 17α-estradiol treatment, we performed stricter screening by intersecting the top 20 and bottom 20 ranks of the scores of c1-up-signature, ferroptosis gene signature, UPR, Mtorc1 signaling, and OXPHOS subunits (*Figure 5A*). Neurons expressing *Calcb*, *Edn3*, *Ucn*, *Ghrl*, *Nmu*, *Npff*, *Cnp*, and *Agrp* ranked among the bottom 20 in at least 4 out of the 5 gene signatures or gene sets. These neurons are involved in stress responses, vascular activity, appetite regulation, and muscle contraction. Notably, the lower levels of *Agrp*- and *Ghrl*-expressing neurons in the Mitocarta_OXPHOS_subunits signature may also indicate reduced physiological activity of these potent appetite-promoting neurons during 17α-estradiol treatment, which could represent a key clue to its role in lifespan prolongation.

In contrast, neurons expressing *Gast*, *Npb*, *Nppa*, *Crh*, *Scg3*, and *Npw* consistently ranked among the top 20 in at least 4 of the 5 gene signatures or gene sets. These neurons participate in gastrointestinal activity, feeding behavior, stress responses, cardio-renal homeostasis, and angiogenesis. Of note, the expression pattern of *Crh* neurons in O.T was opposite to that in O (*Figure 4A*). Additionally, the Mitocarta_OXPHOS_subunits score in *Crh* neurons was the highest among all examined neuropeptide-expressing subtypes (*Figure 5A*), which contrasted sharply with those of *Agrp* and *Ghrl* neurons. Additionally, the treatment with 17α-estradiol in O.T also elevated several key metabolic pathways in *Crh* neurons compared to those in Y and O (*Figure 5B*). 17α-estradiol treatment increased the c1-up-signature while simultaneously reducing many pathways associated with synapse activity and the c4-up-signature in *Crh* neurons of O.T, indicating a potent stressed phenotype in *Crh* neurons. In contrast, in *Kiss1* and *Prlh* neurons, the decreased c1-up-signature in O.T implied a lesser extent of stressed phenotype in these neurons compared to *Crh* neurons. The status of Crh neurons in O.T may be associated with elevated TF activities of *Esr2*, *Usf2*, *Hdac5*, *Creb3l1*, *Tfam*, *Preb*, *Pou3f2*, and *Hoxb5* (*Figure 5C*). The aberrant changes in *Crh* neurons were also evidenced by the increased expression of DEGs related to mitochondria-expressed genes and reduced expression of DEGs in the adherens junction pathway in O.T, indicative of higher energetic activity and altered extracellular adhesion in this type of neuron by 17α-estradiol treatment (*Figure 5D*).

Notably, the HPA axis was altered by 17α-estradiol treatment, as evidenced by the elevated cortisol levels in O.T compared to O (p=0.078) (*Figure 5E*). The correlation between elevated cortisol production and the heightened stress in *Crh* neurons by 17α-estradiol treatment needs further investigation. Additionally, as a crucial component of the renin-angiotensin-aldosterone system, the significantly increased serum aldosterone in O.T and its potential role in sodium reabsorption and cardiovascular health also warrant more detailed investigation (*Figure 5E*).

In summary, 17α-estradiol treatment altered the activity of appetite-promoting neurons and the hypothalamic-pituitary-adrenal (HPA) axis in male BN rats, while also inducing enhanced stress responses in *Crh* neurons.

## 17α-estradiol increased Oxt neuron proportion and secretion and its possible role in mediating the effect of 17α-estradiol on endocrine system

In sample Y, the top four neuropeptide-secreting neuron subclasses, ranked by their proportion, are *Rln1*, *Pomc*, *Npvf*, and *Agrp*; conversely, the bottom 4 are *Oxt*-, *Vip*-, *Avp*-, and *Grp*-secreting neurons (*Figure 6A*). This pattern shows a reciprocal relationship: proportions of *Oxt*-, *Vip*-, and *Avp*-secreting neurons, among others, increase significantly in the O.T sample, with *Oxt*-secreting neurons accounting for the highest proportion. In contrast, the proportions of, *Rln1*-, *Pomc*-, *Npvf*-, and *Agrp* secreting neurons decrease substantially in the O.T sample. Notably, *Agrp* and *Pomc* neurons are well

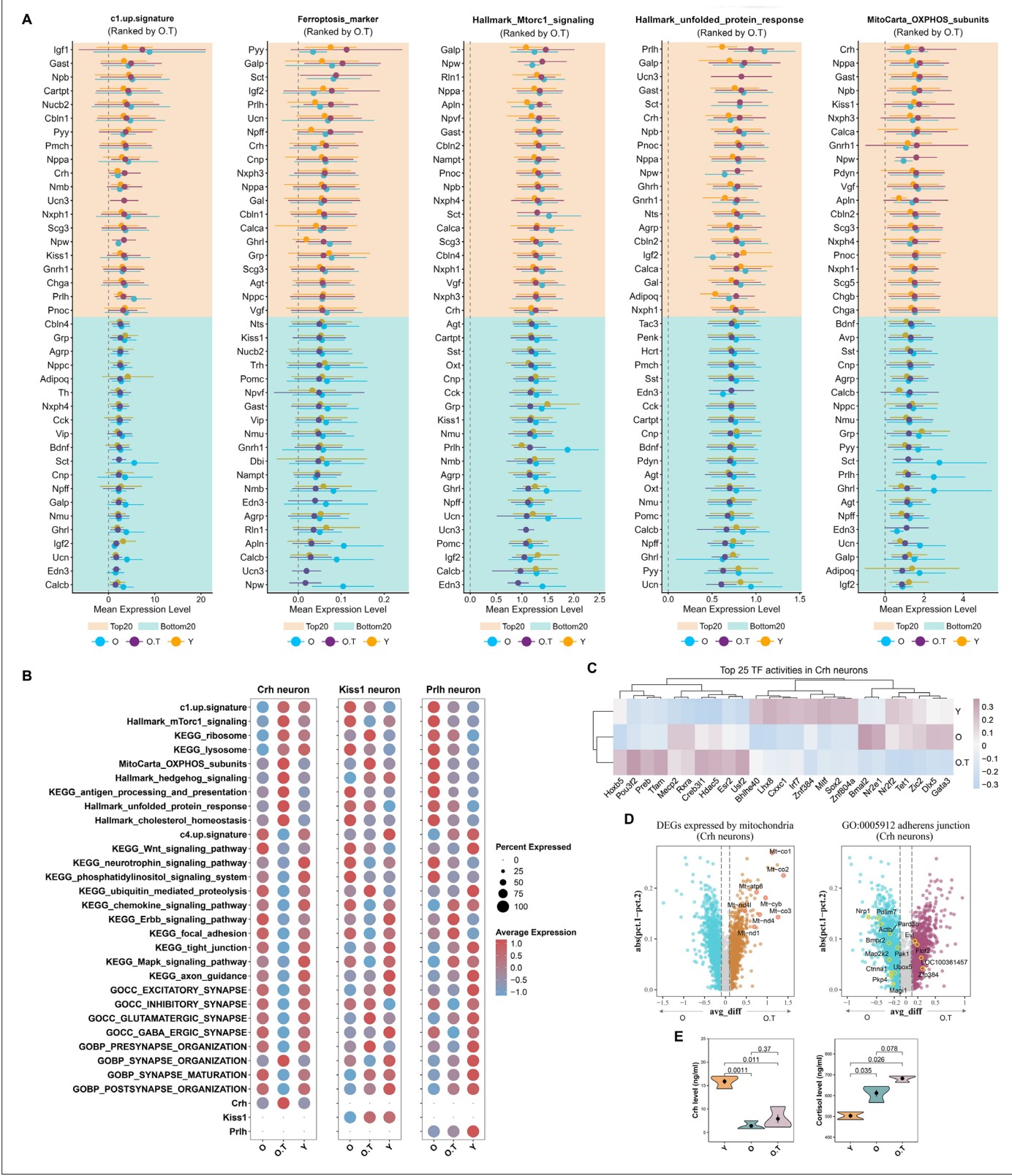

**Figure 5.** Responses of Crh neurons to long-term 17α-estradiol treatment. (**A**) The top 20 and bottom 20 neuropeptide-secreting neuron subtypes, ranked by their mean expression values of five signatures or gene sets in sample O. (**B**) Expression profiles of selected pathways from two opposing signaling networks in *Crh*, *Kiss1*, and *Prlh* neurons. (**C**) Downregulated and upregulated differentially expressed genes (DEGs) associated with mitochondria or the adherens junction pathway in *Crh* neurons, comparing O.T vs O. (**D**) Top 25 transcription factor (TF) activities in *Crh* and *Gnrh1*

*Figure 5 continued on next page*

*Figure 5 continued*

neurons. (**E**) Serum levels of Crh, cortisol, and aldosterone in Y, O, and O.T groups as measured by enzyme immunoassay; two-tailed unpaired t-tests were performed, with p-values indicated.

documented for their roles in regulating food intake and energy homeostasis. Specifically, within the arcuate nucleus (ARC) of the hypothalamus, *Agrp* neurons are activated by hunger, whereas *Pomc* neurons are activated by satiety . In terms of neurons expressing neuropeptide receptors or hormone receptors in sample Y, those with relatively high proportions are *Calcrl*-, *Mc3r*-, *Ednrb*-, and *Ednra*-expressing neurons. On the other hand, the receptor-expressing neurons with relatively low proportions include *Rxfp3*-, *Rxfp2*-, *Mlnr*-, *Sstr2*-, and *Ntsr1*-expressing neurons. 17α-estradiol treatment effectively elevated the expression levels of the c4-up-signature (blue arrows) and synapse-associated processes in neuron subtypes *Agrp*, *Pomc*, *Oxt*, and *Glp2r* in O.T compared to O (*Figure 6B*). This may mitigate the adverse effects of reduced cell populations in *Pomc* and *Agrp* neurons in aging hypothalamus. This finding indicates a potential role of 17α-estradiol in appetite control, as previously reported (*Steyn et al., 2018*). Notably, the proportions of *Oxt* and *Glp2r* neurons, both of which have anorexigenic effects (*Inada et al., 2022*; *Dalvi and Belsham, 2012*), increased in O.T. In addition to the increased number of *Oxt*-positive neurons, the expression level of *Oxt* also rose in O.T. Additionally, 17α-estradiol treatment altered two opposing signaling pathways—those linked to metabolic pathways and synapse-related pathways—in *Agt*, *Vip*, *Avp*, *Npff*, *Calca*, and *Tacr1* neurons (*Figure 6—figure supplement 1*). Notably, all these neuron types are associated with blood pressure regulation.

In addition to the increased number of *Oxt*-positive neurons, the expression level of *Oxt* also rose in O.T (*Figure 6B*). The elevated expression of synapse-related pathways was supported by the increased DEGs in the enriched synaptic membrane pathway in *Oxt* neurons (*Figure 6C*). More importantly, the serum level of Oxt was significantly elevated in O.T compared to O (p=0.04), yet remained lower than those in Y (*Figure 6D*). Notably, the top TF activities in O.T and O differed markedly from those in Y (*Figure 6E*). The elevated levels of *Hopx* and *Xbp1* may be associated with the response to 17α-estradiol treatment.

Due to the intricate regulatory networks among various endocrine factors, elucidating the causal effect of Oxt on other endocrine factors is quite complex using traditional methods. MR analysis, employing variant SNPs as genetic tools, is advantageous for such tasks. We performed a bidirectional MR analysis of the GWAS summary data of human plasma OXT and 204 endocrine-related and hypothalamus-related factors, most of which are protein quantitative trait loci (pQTL) data from the IEU (*Supplementary file 3*). As an exposure, OXT revealed a significant causal effect on POMC/beta-endorphin (id:prot-a-2327, id:prot-a-2325), glucagon (id:prot-a-1181), GNRH1/Progonadoliberin-1 (id:prot-a-1233), and total testosterone levels (id:ebi−a−GCST90012112, id:ieu−b−4864) (*Figure 6F*). NPW and CBLN1 were found to be negatively associated with OXT, but the significance of these associations was not found in the reverse MR analysis (*Figure 6—figure supplement 2A, B*).

In contrast, we could not identify significant associations between OXT and estradiol levels (id:ebi-a-GCST90012105, id:ebi-a-GCST90020092, id:ebi-a-GCST90020091, id:ieu-b-4872, id:ieu-b-4873, id:ukb-e-30800_AFR, id:ukb-e-30800_CSA). Interestingly, QRFP, IGF1, AGRP, TAC4, GRP, CLU, BNF, PCSK7, PACAP, ANP, TAC3, CRH, INSL6, and PRL displayed significant associations with OXT in both MR and reverse MR analysis, indicative of their complex causal effects (*Figure 6—figure supplement 2A and B*).

The results suggest that elevated Oxt levels induced by 17α-estradiol may have positive associations with endocrine factors governing feeding behavior, glucose metabolism, male reproduction, and sex hormones. Therefore, OXT may serve as a potential mediator of 17α-estradiol.

## 17α-estradiol activated HPG axis and the elevated Gnrh also took important roles in mediating the effect of 17α-estradiol on other endocrine factors

Given the sensitivity of GnRH- and sex hormone receptor-expressing neuron subtypes to 17α-estradiol treatment (*Figure 3C and D*), we analyzed their expression profiles alongside representative pathway genes from two opposing signaling networks - those related to metabolism and synapses

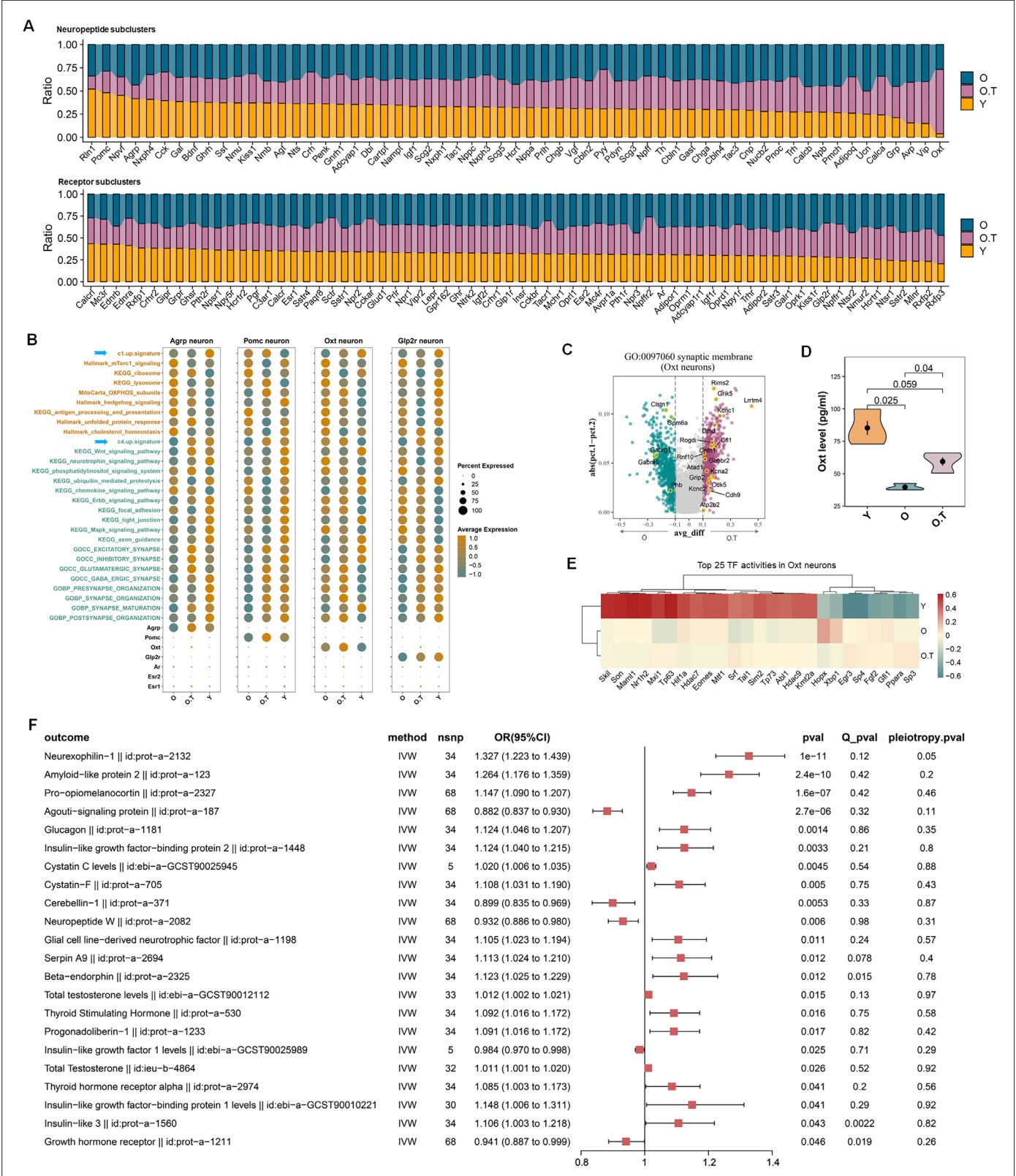

**Figure 6.** The response of oxytocin (*Oxt*) neurons to 17α-estradiol and the causal effects of Oxt on other endocrine factors. (**A**) The relative cell proportions of peptide-expressing subclusters (upper panel) and receptor-expressing subclusters (lower panel) across Y, O, and O.T (sorted in descending order of proportions in Y). Only subclusters with a cell count of n≥10 in sample Y were included for calculation. (**B**) Dot plots showing the expression profiles of the selected pathways from the two opposing signaling pathways in four types of food uptake-related neurons, which decreased

*Figure 6 continued on next page*

*Figure 6 continued*

or increased among the top 10 ranks in (**A**) or (**B**). Blue arrows: c1-up-signature and c4-up-signature. (**C**) Volcanic plots showing the differentially expressed genes (DEGs) between Neuron.O.T and Neuron.O in the pathway synaptic membrane. (**D**) Enzyme immunoassay of the plasma levels of Oxt in three groups. (**E**) Top 25 transcription factor (TF) activities in neuron Oxt. (**F**) Significant causal effects (p<0.05, inverse-variance weighting IVW) between exposure OXT (id: prot-a-2159) and 204 endocrine-related outcomes, which were not significant in reverse Mendelian randomization (MR) analysis. Significant heterogeneity (Q_pval <0.05). Significant horizontal pleiotropy (pval <0.05).

The online version of this article includes the following figure supplement(s) for figure 6:

**Figure supplement 1.** The expression profiles of selected pathways from the two opposing signaling networks in 6 cardiovascular system-related neurons.

**Figure supplement 2.** Bidirectional two-sample MR analysis of causal effects between 203 endocrine-related factors and Oxt (id: prot-a-2159).

- such as the c1-up-signature and c4-up-signature (*Figure 7A*). However, neither the c1-up-signature nor the c4-up-signature was up-regulated in *Gnrh1* neuron in the O.T in comparison with Y. *Ar* and *Esr2* neuron displayed decreased level of c1-up-signature in comparison with O. Only in *Esr1* neuron was the c1-up-signature found to be up-regulated. Meanwhile, both *Ar* and *Esr* neurons displayed increased level of c4-up-signature in O.T comparing with O. *Ar*, *Pgr*, and *Esr1* were also among the top 20 of c4-up-signature, suggesting long-term 17α-estradiol treatment did not impose significant stress on hypothalamic neurons expressing these hormone receptors (*Figure 7—figure supplement 1*). But *Gnrh1* and *Crh* neurons were among the bottom 20, indicative of higher cellular stress by long-term 17α-estradiol treatment. However, based on these cellular perturbations, it's difficult to define the precise physiological status of these subtypes of neurons, particularly regarding neuroendocrine activities. Consequently, we performed enzyme immunoassays of hormones from the serum of O, O.T, and Y. The treatment with 17α-estradiol significantly increased the plasma level of Gnrh compared to Y (p=0.0099) and approached significance when compared to O (p=0.096) (*Figure 7B*). More intriguingly, testosterone levels in serum were significantly increased in O.T compared to O (p=0.018) and approached significance when compared to Y (p=0.052). Additionally, the serum estradiol levels were significantly increased in O compared to Y (p=0.011) and significantly decreased in O.T compared to O (p=0.019), suggesting that 17α-estradiol treatment markedly altered the homeostasis of testosterone and estradiol.

Furthermore, most testes from 30-month-old male BN rats exhibited severe age-related inflammation and epithelial collapse of seminiferous tubules (*Figure 7C*). The testes without inflammation in O.T displayed normal morphology. 17α-estradiol treatment slightly decreased the testis inflammation in O.T compared to that in O (p=0.15), indicating a potential positive role of 17α-estradiol treatment in male reproductive system. The elevated TFs such as *Sf1*, *Pparg*, *Litaf*, *Nupr1*, *Rxrg*, *E2f2*, and *Zfp42* may be involved in the transcriptional regulation by 17α-estradiol in O.T (*Figure 7D*). Importantly, the activities of androgen and estrogen pathways were decreased in *Gnrh1* neurons in O.T compared to O, and were distinct from those in *Ar*, *Esr1*, and *Esr2* neurons (*Figure 7E*). These signaling pathways are important for the feedback control of sex hormone secretion in *Gnrh* neurons, and these results may also reflect the strong effect of 17α-estradiol on *Gnrh* neurons.

To decipher the potential effects of elevated serum Gnrh levels on endocrine system, we performed bidirectional MR analysis of the GWAS summary data of human GNRH1 (id: prot-a-1233) and 204 endocrine-related factors with genetic variants SNPs. We found strong causal effects of GNRH1 on GAL/Galanin (id:prot−a−1166), POMC/Beta−endorphin (id:prot−a−2327, id:prot−a−2325), Adrenomedullin (id:prot−a−48), BDNF (id:prot−a−242)**,** and LPR (id:prot−a−1724), which are involved in feeding, energy homeostasis, osmotic regulation, and neuron plasticity (*Figure 7F*). Notably, CRH/Corticotropin (id:prot−a−2326), PRLH/Prolactin−releasing peptide (id:prot−a−2376), NPW/Neuropeptide W (id:prot−a−2082), Glucagon (id:prot−a−1181), Chromogranin−A (id:prot−a−538) displayed bidirectional significance, indicating close and complex causal effects between GNRH1 and these endocrine factors (*Figure 7G*, *Figure 7—figure supplement 2A and B*). These results also suggest that the role of 17α-estradiol treatment in feeding, energy homeostasis, reproduction, osmotic regulation, stress response, and neuronal plasticity may be mediated, at least in part, by elevated Gnrh secretion.

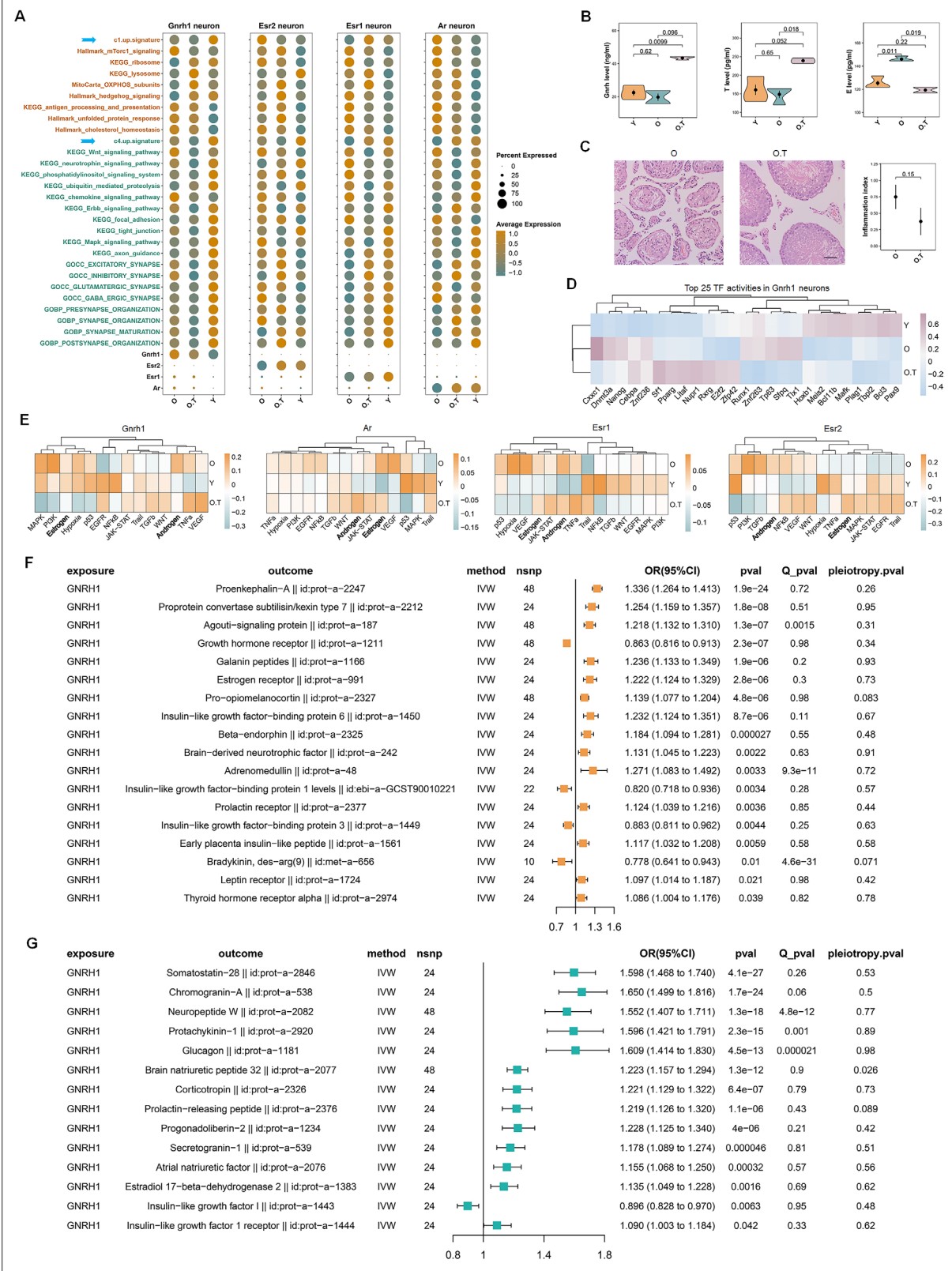

**Figure 7.** The response of hypothalamic-pituitary-gonadal (HPG) axis in males to 17α-estradiol and the causal effects of gonadotropin-releasing hormone (Gnrh) on other endocrine factors. (**A**) The expression profiles of pathways from the two opposing signaling networks in *Gnrh1*-, *Esr2*-, *Esr1*-, or *Ar*-positive neurons. (**B**) Enzyme immunoassay of the serum levels of Gnrh, total testosterone (T), and estrogen (E) in Y, O, and O.T samples. Two-tailed unpaired t-test was performed. (**C**) Inflammation of seminiferous tubules in testes of O and O.T. Left two panels: representative HE staining of

*Figure 7 continued on next page*

*Figure 7 continued*

testis inflammation in O and the normal seminiferous tubules of O.T. Right panel: the mean testis inflammation index of O and O.T. Bar, 50 μm (**D**) The top 25 TF activities in *Gnrh1* neurons in three groups. (**E**) The activities of 14 pathways in *Gnrh1-*, *Esr2-*, *Esr1-*, or *Ar*-positive neurons. (**F**) Significant causal effects (inverse-variance weighting IVW, p<0.05) between exposure GNRH1 (id: prot-a-1233) and 204 endocrine-related outcomes, which were not significant in reverse MR analysis. (**G**) Items with significant causal effects (IVW, p<0.05) in both directions of MR analysis between GNRH1 (id: prot-a-1233) and 204 endocrine-related outcomes.

The online version of this article includes the following figure supplement(s) for figure 7:

**Figure supplement 1.** The top 20 and bottom 20 neuron subtypes based on the mean expression values of c4-up-signature, ranked by the values in sample O.T, in neuropeptide-, or hormone-secreting subtypes (upper panel) and in neuron subtypes expressing neuropeptide receptors or hormone receptors (lower panel).

**Figure supplement 2.** Two-sample MR analysis of causal effects of 204 endocrine-related exposures on outcome GNRH1.

## Discussion

The most striking effect of 17α-estradiol treatment revealed in this study is its modulation of the HPG axis: serum levels of Gnrh and testosterone were significantly elevated in the O.T group compared to the O group, which may counteract age-related declines in HPG axis activity. The underlying molecular mechanism remains unclear; however, prior reports have indicated that 17α-estradiol can bind to ESR1 (***Mann et al., 2020***). In our findings, 17α-estradiol treatment significantly decreased serum estradiol levels while elevating serum testosterone. Based on this evidence, we propose that 17α-estradiol may function similarly to estrogen receptor antagonists or aromatase inhibitors, potentially preventing the conversion of androgens to estrogens (***Leder et al., 2004***; ***Guay et al., 2003***). These actions could alleviate the feedback inhibition exerted by estrogen on hypothalamus and pituitary, thereby facilitating the secretion of Gnrh, FSH, and LH (***Wang and Swerdloff, 2022***).

The testosterone levels in men gradually decline beginning in the third decade of life (***Camacho et al., 2013***). Age-related deterioration of the gonadotropic axis, particularly in older males with low serum testosterone, is often linked to numerous aging symptoms, including loss of skeletal muscle mass, reduced muscle strength and power, low bone mineral density, frailty, impaired physical performance, mobility limitations, increased risk of diabetes, elevated all-cause cardiovascular mortality, cognitive decline, and heightened risk of Alzheimer's disease (***Rodrigues Dos Santos and Bhasin, 2021***). Consequently, testosterone supplementation in older men is beneficial. Additionally, Gnrh supplementation may help mitigate age-related declines in neurogenesis and slow aging processes (***Zhang et al., 2013***). Importantly, treatment with 17α-estradiol did not result in feminization or adversely affect the sperm parameters and fertility in male animals (***Stout et al., 2017***; ***Isola et al., 2023***). Thus, the observed increases in Gnrh and serum testosterone levels due to 17α-estradiol treatment are likely advantageous for older males, particularly those experiencing late-onset hypogonadism.

Postmenopausal women with low estrogen experience aging-related syndromes similar to those of older males with low serum testosterone. Those women also face increased mortality, cardiovascular disease, osteoporosis fracture, urogenital atrophy, and dementia, all of which may benefit from hormone therapy (***Shoupe, 2011***). However, a prior report indicates that 17α-estradiol treatment does not provide positive life extension effects in aged females (***Harrison et al., 2014***). The discrepancy may stem from the inhibitory effects of estrogens associated with 17α-estradiol treatment, as evidenced by its inability to enhance female fertility (***Isola et al., 2022***). Nonetheless, due to the lack of parallel data in aged female BN rats treated with 17α-estradiol, further research is needed to definitely address this question in the female subjects in the future.

The stressed phenotype observed in neuronal subtypes discussed herein likely represents a transcriptomic manifestation of heightened physiological activity. For instance, as evidenced in this study, prolonged 17α-estradiol treatment induces a pronounced stressed phenotype in *Gnrh* neurons alongside elevated Gnrh secretion and consequent high serum testosterone levels. Similarly, the stressed phenotype reflected in the *Crh* neuronal transcriptome coincides with substantially increased serum cortisol. Furthermore, long-term 17α-estradiol treatment markedly alleviates the stressed phenotype in appetite-stimulating neurons (*Agrp* and *Ghrl*), suggesting an appetite-suppressing effect in rats. This aligns with previously reported findings that 17α-estradiol treatment inhibits feeding behavior in mice, as Agrp and Ghrl neurons are key promoters of appetite; reduced stress (and potentially

reduced activity) in these neurons in O.T may weaken appetite drive, contributing to inhibited feeding (*Steyn et al., 2018*).

Another notable effect of 17α-estradiol is its ability to reduce the overall expression levels of energy metabolism in hypothalamic neurons of aged male BN rats. The nutrient-sensing network, mediated by MTORC1 complex, is a central regulator of mRNA and ribosome biogenesis, protein synthesis, glucose metabolism, autophagy, lipid metabolism, mitochondrial biosynthesis, and prote-asomal activity (*López-Otín et al., 2023*). Downregulation of this nutrient-sensing network has been associated with increased lifespan and healthspan (*Singh et al., 2019*). Notably, 17α-estradiol treat-ment diminished nutrient-sensing network activity in most hypothalamic neurons, which may be a contributing factor in promoting lifespan extension.

In this report, we demonstrated significant changes in neuron populations involved in appetite control, including *Agrp*, *Pomc*, *Oxt*, and *Glp2r* neurons. Among the identified subtypes, the propor-tion of *Oxt* neurons saw the most considerable increase due to 17α-estradiol treatment (*Figure 6A*). Oxt plays versatile roles in social behavior, stress response, satiety, energy balance, reproduction, and inflammation (*Kerem and Lawson, 2021*). Most *Oxt* neurons originate from the paraventric-ular nucleus (PVN) and supraoptic nuclei (SON) in the hypothalamus, exhibiting high plasticity during development and intricate circuitry (*Rosen et al., 2008*; *Madrigal and Jurado, 2021*). The PVN, ARC, and ventromedial hypothalamic nucleus together form a neural hub in the hypothalamus that integrates peripheral, nutritional, and metabolic signals to regulate food intake and energy balance (*Cornejo et al., 2016*). Many effects of Oxt are sex-specific *Carvalho Silva et al., 2023*; for instance, females are less sensitive to exogenous Oxt than males regarding social recognition (*Dumais and Veenema, 2016*). Interestingly, Oxt injections, facilitated by nanoparticles that enhance blood-brain barrier penetration, reduced body mass while increasing social investigation and the number of Oxt-positive cells in the SON, particularly in male rats (*Duarte-Guterman et al., 2020*). Additionally, intra-cerebroventricular injections of Oxt in rats showed a reduction in food intake in both sexes, with a more pronounced effect in males (*Liu et al., 2020*). Therefore, we propose that Oxt's role in systemic aging and feeding behavior may contribute to the sex-biased effects of 17α-estradiol. This hypoth-esis warrants further verification, such as investigating Oxt signaling in female models treated with 17α-estradiol.

Furthermore, 17α-estradiol treatment appears to have enhanced stress in HPA axis. One evidence was the increased levels of ferroptosis signature and UPR in *Crh* neurons. The other evidence was the elevated serum cortisol, which is also a potential hallmark of aging HPA axis (*Veldhuis, 2013*; *Warde et al., 2023*). Therefore, more attention should be paid to the potential side effects of 17α-estradiol, especially in its clinical application.

In summary, our findings suggest that 17α-estradiol treatment positively influences the HPG axis and neurons associated with appetite and energy balance. This may be closely linked to the life-extension effects of 17α-estradiol in aged males. Additionally, employing supervised clustering based on neuropeptides, hormones, and their receptors proves to be a valuable strategy for examining pharmacological, pathological, and physiological processes in different neuronal subtypes within the hypothalamus.

# Materials and methods
## Animals, treatment, and tissues

Twelve Norway brown male rats (12-months-old) were acquired from Charles River, including 8 12- months-old and 4 1-month-old (Beijing). Older rats (12-months-old) were randomly allocated into control and 17α-estradiol-treated groups. Four aged rats treated with 17α-estradiol (Catalog #: E834897, Macklin Biochemical, Shanghai, China) were fed freely with regular diet mixed with 17α-es-tradiol at a dose of 14.4 mg/kg (14.4 ppm), starting at 24 months of age for 6 months according to prior reports (*Harrison et al., 2021*; *Strong et al., 2016*). The young rats were fed a regular diet without 17α-estradiol continuously for 3 months until 4 months old. All rats had ad libitum access to food and water throughout the experiments. The rats were then euthanized via $CO_2$, hypothalami, testes, and blood serum were collected for subsequent experimental procedures. All blood samples were collected at 9:00-9:30 a.m to minimize hormone fluctuation between animals. All animal procedures

were reviewed and approved by the Institutional Animal Care and Use Committee at Nantong University (approval number: S20210225-012).

## Enzyme immunoassays

Enzyme immunoassay kits for rat Oxt (Catalog #: EIAR-OXT), Corticotropin Releasing Factor (Catalog #: EIAR-CRF), and gonadoliberin-1 (Catalog #: EIAR-GNRH) were obtained from Raybiotech (GA, USA). Enzyme immunoassay kits for rat serum total testosterone (Catalog #: ml002868), estradiol (Catalog #: ml002891), aldosterone (Catalog #: ml002876), and cortisol (Catalog #: ml002874) were obtained from Enzyme-linked Biotechnology (Shanghai, China). Sera from three animals per group were used and each was diluted 10 or 20 times for immunoassays.

## Seminiferous tubule inflammation test

Eight testes were obtained from each sample group and then subjected to fixation in 4% formalin for at least 1 week. Formalin-fixed paraffin-embedded rat testis sections of 5 μm thickness were used for HE staining. At least 30 seminiferous tubules in each slide were examined for inflammation tests. Testis with at least 1 inflammatory seminiferous tubule was set as 1, and normal testis was set as 0 for inflammation index calculation.

## snRNA-seq data processing, batch effect correction, and cell subset annotation

Intact hypothalami were cryopreserved in liquid nitrogen from sacrificed rats. Two (O) or three (Y and O.T) hypothalami were pooled within each group and homogenized in 500 μL ice-cold homogenization buffer (0.25 M sucrose, 5 mM $CaCl_2$, 3 mM $MgAc_2$, 10 mM Tris-HCl [pH 8.0], 1 mM DTT, 0.1 mM EDTA, 1× protease inhibitor, and 1 U/μL RiboLock RNase inhibitor) with Dounce homogenizer. Then, the homogenizer was washed with 700 μL ice-cold nuclei washing buffer (0.04% bovine serum albumin, 0.2 U/μL RiboLock RNase Inhibitor, 500 mM mannitol, 0.1 mM phenylmethanesulfonyl fluoride protease inhibitor in 1× phosphate buffer saline). Next, the homogenates were filtered through a 70 μm cell strainer to collect the nuclear fraction. The nuclear fraction was mixed with an equal volume of 50% iodixanol and added on top of a 30% and 33% iodixanol gradient. This solution was then centrifuged for 20 min at 10,000×g at 4 °C. After the myelin layer was removed from the top of the gradient, the nuclei were collected from the 30% and 33% iodixanol interface. The nuclei were resuspended in nuclear wash buffer and resuspension buffer and pelleted for 5 min at 500×g at 4 °C. The nuclei were filtered through a 40 μm cell strainer to remove cell debris and large clumps, and the nuclear concentration was manually assessed using trypan blue counterstaining and a hemocytometer. Finally, the nuclei were adjusted to 700–1200 nuclei/μL, and examined with a 10 X Chromium platform.

Reverse transcription, cDNA amplification, and library preparation were performed according to the protocol from 10 X Genomics and Chromium Next GEM Single Cell 3′ Reagent Kits v3.1. Library sequencing was performed on the Illumina HiSeq 4000 by Gene Denovo Biotechnology Co., Ltd (Guangzhou, China).

10 X Genomics Cell Ranger software (version 3.1.0) was used to convert raw BCL files to FASTQ files, and for alignment and counts quantification. Reads with low-quality barcodes and UMIs were filtered out and then mapped to the reference genome. Reads uniquely mapped to the transcriptome and intersecting an exon at least 50% were considered for UMI counting. Before quantification, the UMI sequences were corrected for sequencing errors, and valid barcodes were identified using the EmptyDrops method. The cell ×gene matrices were produced via UMI counting and cell barcodes calling. Cells with an unusually high number of UMIs (≥8000) or mitochondrial gene percent (≥15%) were filtered out. Batch effect correction was performed by harmony.

## Pathways, gene signatures, TFs and TF cofactors, cell communication

Gene sets and pathways were derived from Hallmark gene sets of MSigDB collections, the KEGG pathway database, Reactome pathway database, and WikiPathways database, and some ontology terms derived from the Gene Ontology (GO) resource. Mitochondrial pathways were derived from MitoCarta3.0 (*Rath et al., 2021*). Pathways, gene sets, and gene signatures were evaluated with the

PercentageFeatureSet function built into R package Seurat. TFs and TF cofactors were obtained from AnimalTFDB 3.0 (*Hu et al., 2019*). TFs and TF cofactors were further filtered with mean counts >0.1. The ligand–receptor pairs were calculated via R package CommPath (*Lu et al., 2022*).

## Correlation analysis and ROC analysis

Pearson correlation coefficient was calculated with the linkET package (p<0.05). A total of 431 pathways from Hallmark, KEGG, and PID databases were used for correlation analysis with MitoCarta OXPHOS subunits in neurons and non-neural cells and the top 20 and bottom 20 items according to the correlation coefficient values in Neuron.O were shown (*Figure 2—figure supplement 1*). Fast Wilcoxon rank sum test and auROC analysis was performed with the wilcoxauc function in R package presto. The minimal cell number in either one of the comparing pairs should be no less than 15. Ranks of area under the curve (AUC) values were in descending order. A total of 97 pathways related to synapse activity were derived from GO, including GO cellular components, GO biological processes, and GO molecular functions (*Supplementary file 1*).

## The division of expression level-dependent clusters in each pathway and their gene signatures

The quarters of the mixed cell populations from O, O.T, and Y hypothalamic neurons were equally divided using the R function fivenum from the R package stats, based on pathway expression levels. Thus, the total number of neurons was evenly divided into four clusters (c1-c4) in terms of cell number. The cell proportions from O.T, O, and Y neurons in each cluster were weighted against the total number of neurons in the three groups. The unique markers of each cluster were calculated using the FindAllMarkers function from the Seurat package. The intersection of the unique markers from the six pathways was obtained for heatmap plotting. Nineteen genes that were highly expressed in c1 were identified as c1.up.signature via the PercentageFeatureSet function in the Seurat package. Twelve genes that were highly expressed in c4 were identified as c4.up.signature. There were no intersecting unique markers in clusters c2 and c3 among the six selected pathways.

## TF and pathway activities

The TF resources were derived from CollecTRI, the pathway resource was from PROGENy, and the enrichment scores of TFs and pathways were performed with the Univariate Linear Model (ulm) method according to the pipeline in R package decoupleR (*Badia-i-Mompel et al., 2022*).

## Subtypes of neurons generated by supervised clustering and cell prioritization

Vast majority of these subtypes were clustered by neuropeptides, hormones, and their receptors within all the neurons with the subset function from R package Seurat (the target gene expression level >0). A total of 209 neuron subtypes were obtained, comprising 104 neuropeptide-secreting or hormone-secreting neurons and 105 neurons expressing a unique neuropeptide receptor or hormone receptor (*Supplementary file 2*). Further groupings may exist within the identified neuron subtypes, and the category of excitatory or inhibitory neurons was not discriminated further. The cell proportion of each neuron subtype was weighted according to the total number of neurons in O.T, O, and Y samples. The mean values ± standard deviation of pathways and gene signatures were performed for each subtype. The top 20 and the bottom 20 items were calculated. The cell type prioritization was performed using the R package Augur, with the subsample_size parameter of the calculate_auc function set to 6 (*Skinnider et al., 2021*). In each comparison pair, the minimum number of cells in a subcluster shall not be less than 6 when performing cell prioritization with function calculate_auc.

## Differential expression and pathway enrichment analysis

DEGs between groups were identified via FindMarkers (test.use=bimod, min.pct=0.1, logfc. Threshold=0.25, avg_diff >0.1 or < −0.1). DEGs were then enriched in redundant GO terms via WebGestalt and filtered with false discovery rate <0.05 (*Liao et al., 2019*).

## Bidirectional MR study

The protein quantitative trait locus (pQTL) data of 204 human endocrine-related GWAS summary data with European ancestry were obtained from open-access MRC Integrative Epidemiology Unit (IEU)

(*Supplementary file 3*; *Hemani et al., 2018*; *Sun et al., 2018*). Independent genome-wide significant SNPs for exposure OXT (id:prot-a-2159) or GNRH1 (id: prot-a-1233) were used as instrumental variables with genome-wide significance ($p < 1 \times 10^{-5}$), independence inheritance (r2 <0.001) without linkage disequilibrium (LD) with each other for MR. For the reverse MR, independent genome-wide significant SNPs from 204 endocrine-related GWAS summary data ($p < 1 \times 10^{-5}$, r2 <0.001) without LD with each other were obtained as exposures and human GWAS summary data of OXT (id:prot-a-2159) or GNRH1 (id:prot-a-1233) were used as outcomes. Weak instruments less than 10 were discarded via F-statistics.

MR and reverse MR analysis were conducted with method inverse-variance weighting (IVW), MR Egger, Weighted median, Simple mode, and Weighted mode. The screening criteria: all of the odds ratio (OR) values of the 5 methods should be simultaneously either >1 or <1 and the significant p-value of IVW was <0.05. The heterogeneity via IVW method and the horizontal pleiotropy were also evaluated with R package TwoSampleMR (*Hemani et al., 2017*).

## Acknowledgements

We appreciate Dr. Qinghua Wang and Hongyun Shi from animal facility of Nantong University in helping with the animal experiments. Special thanks to Professor Ken-ichiro Fukuchi from the University of Illinois College of Medicine for constructive comments and suggestions in manuscript preparation. This work was supported by the National Natural Science Foundation of China grant 31271448 (YL), 82171621 (LL), 82172566 (ZY), and the National High Level Hospital Clinical Research Funding (2022-PUMCH-A-231) (LL).

## Additional information

### Funding

| Funder | Grant reference number | Author |
| --- | --- | --- |
| National Natural Science Foundation of China | 31271448 | Yinchuan Li |
| National Natural Science Foundation of China | 82171621 | Lei Li |
| National Natural Science Foundation of China | 82172566 | Zhuo Yu |
| National High Level Hospital Clinical Research Funding | 2022-PUMCH-A-231 | Lei Li |

The funders had no role in study design, data collection and interpretation, or the decision to submit the work for publication.

### Author contributions

Lei Li, Conceptualization, Funding acquisition, Investigation, Project administration, Writing – review and editing; Guanghao Wu, Resources, Methodology; Xiaolei Xu, Data curation, Methodology; Junling Yang, Formal analysis, Validation, Writing – review and editing; Lirong Yi, Investigation; Ziqing Yang, Resources, Validation, Writing – review and editing; Zheng Mo, Resources, Software; Li Xing, Resources; Ying Shan, Resources, Supervision, Investigation, Writing – review and editing; Zhuo Yu, Supervision, Funding acquisition, Validation, Project administration; Yinchuan Li, Conceptualization, Formal analysis, Investigation, Visualization, Methodology, Writing – original draft, Project administration

### Author ORCIDs

Yinchuan Li ⓘ https://orcid.org/0000-0002-8450-6126

### Ethics

All animal procedures were reviewed and approved by the Institutional Animal Care and Use Committee at Nantong University (permit Number S20210225-012).

Reviewer #1 (Public review): https://doi.org/10.7554/eLife.100346.4.sa1
Reviewer #2 (Public review): https://doi.org/10.7554/eLife.100346.4.sa2
Author response https://doi.org/10.7554/eLife.100346.4.sa3

## Additional files

### Supplementary files

Supplementary file 1. auROC Analysis of 97 Synapse Activity-Related Pathways.

Supplementary file 2. List of Neuropeptides, Receptors, and Signatures.

Supplementary file 3. Protein Quantitative Trait Locus (pQTL) Data of 204 Human Endocrine-Related GWAS Summary Datasets.

MDAR checklist

### Data availability

All sequencing data are available at GEO accession number GSE248413.

The following dataset was generated:

| Author(s) | Year | Dataset title | Dataset URL | Database and Identifier |
|---|---|---|---|---|
| Li Y | 2024 | Single-nucleus transcriptomic sequencing of aging hypothalamus treated with 17a-estradiol | https://www.ncbi.nlm.nih.gov/geo/query/acc.cgi?acc=GSE248413 | NCBI Gene Expression Omnibus, GSE248413 |

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
