## [Editor Report · eLife Assessment]

This study demonstrates the potential role of 17α-estradiol in modulating neuronal gene expression in the aged hypothalamus of male rats, identifying key pathways and neuron subtypes affected by the drug. While the findings are **useful** and provide a foundation for future research, the strength of supporting evidence is **incomplete** due to the lack of female comparison, a young male control group, unclear link to 17α-estradiol lifespan extension in rats, and insufficient analysis of glial cells and cellular stress in CRH neurons.

---

## [Referee Report · Reviewer #1 (Public review)]

Summary:

Previous studies have shown that treatment with 17α-estradiol (a stereoisomer of the 17β-estradiol) extends lifespan in male mice but not in females. The current study by Li et al, aimed to identify cell-specific clusters and populations in the hypothalamus of aged male rats treated with 17α-estradiol (treated for 6 months). This study identifies genes and pathways affected by 17α-estradiol in the aged hypothalamus.

Strengths:

Using single-nucleus transcriptomic sequencing (snRNA-seq) on hypothalamus from aged male rats treated with 17α-estradiol they show that 17α-estradiol significantly attenuated age-related increases in cellular metabolism, stress, and decreased synaptic activity in neurons.

Moreover, sc-analysis identified GnRH as one of the key mediators of 17α-estradiol's effects on energy homeostasis. Furthermore, they show that CRH neurons exhibited a senescent phenotype, suggesting a potential side effect of the 17α-estradiol. These conclusions are supported by supervised clustering by neuropeptides, hormones, and their receptors.

Weaknesses:

However, the study has several limitations that reduce the strength of the key claims in the manuscript. In particular:

(1) The study focused only on males and did not include comparisons with females. However, previous studies have shown that 17α-estradiol extends lifespan in a sex-specific manner in mice, affecting males but not females. Without the comparison with the female data, it's difficult to assess its relevance to the lifespan.

(2) Its not known whether 17α-estradiol leads to lifespan extension in male rats similar to male mice. Therefore, it is not possible to conclude that the observed effects in the hypothalamus, are linked to the lifespan extension. The manuscript cited in the introduction does not include lifespan data on rats.

(3) The effect of 17α-estradiol on non-neuronal cells such as microglia and astrocytes is not well described (Fig.1). Previous studies demonstrated that 17α-estradiol reduces microgliosis and astrogliosis in the hypothalamus of aged male mice. Current data suggest that the proportion of oligo, and microglia were increased by the drug treatment, while the proportions of astrocytes were decreased. These data might suggest possible species differences, differences in the treatment regimen, or differences in drug efficiency. This has to be discussed.

A more detailed analysis of glial cell types within the hypothalamus in response to drug should be provided.

(4) The conclusion that CRH neurons are going into senescence is not clearly supported by the data. A more detailed analysis of the hypothalamus such as histological examination to assess cellular senescence markers in CRH neurons, is needed to support this claim.

Revised submission:

Some of the concerns were addressed in this revised version, and the authors responded and addressed study design limitations in both sexes/ages.

However, there are still some concerns that were not sufficiently addressed:

While the term "senescent" was changed to "stressed," some histological/ cellular validation of this phenotype is still needed.

Some discussion on the sex-specific effects of 17α-estradiol in the hypothalamus is still required. Previous studies in mice demonstrated that 17α-estradiol reduced hypothalamic microgliosis and astrogliosis in male but not female UM-HET3 mice.

Additionally, the provided analysis on astrocytes and microglia is superficial.

---

## [Referee Report · Reviewer #2 (Public review)]

Summary:

Li et al. investigated the potential anti-ageing role of 17α-Estradiol on the hypothalamus of aged rats. To achieve this, they employed a very sophisticated method for single-cell genomic analysis that allowed them to analyze effects on various groups of neurons and non-neuronal cells. They were able to sub-categorize neurons according to their capacity to produce specific neurotransmitters, receptors, or hormones. They found that 17α-Estradiol treatment led to an improvement in several factors related to metabolism and synaptic transmission by bringing the expression levels of many of the genes of these pathways closer or to the same levels to those of young rats, reversing the ageing effect. Interestingly, among all neuronal groups, the proportion of Oxytocin-expressing neurons seems to be the one most significantly changing after treatment with 17α-Estradiol, suggesting an important role of these neurons on mediating its anti-ageing effects. This was also supported by an increase in circulating levels of oxytocin. It was also found that gene expression of corticotropin-releasing hormone neurons was significantly impacted by 17α-Estradiol even though it was not different between aged and young rats, suggesting that these neurons could be responsible for side effects related to this treatment. This article revealed some potential targets that should be further investigated in future studies regarding the role of 17α-Estradiol treatment in aged males.

Strengths:

• The single nucleus mRNA sequencing is a very powerful method for gene expression analysis and clustering. The supervised clustering of neurons was very helpful in revealing otherwise invisible differences between neuronal groups and helped identify specific neuronal populations as targets.

• There is a variety of functions used that allowed the differential analysis of a very complex type of data. This led to a better comparison between the different groups in many levels.

• There were some physiological parameters measured such as circulating hormone levels that helped the interpretation of the effects of the changes in hypothalamic gene expression.

Weaknesses:

• One main control group is missing from the study, the young males treated with 17α-Estradiol.

• Even though the technical approach is a sophisticated one, analyzing the whole rat hypothalamus instead of specific nuclei or subregions makes the study weaker.

• Although the authors claim to have several findings, the data fail to support these claims.

• The study is about improving ageing but no physiological data from the study demonstrated such claim with the exception of the testes histology which was not properly analyzed and was not even significantly different between the groups.

• Overall, the study remains descriptive with no physiological data to demonstrate that any of the effects on hypothalamic gene expression is related to metabolic, synaptic or other function.

Comments on revisions:

The authors revised part of the manuscript to address some of the reviewers' comments. This improved the language and the text flow to a certain extent. They also added an additional analysis including glial cells. However, they failed to address the main weaknesses brought up by the reviewers and did not add any experimental demonstration of their claims on lifespan expansion induced by 17α-estradiol in rats (the cited study does not include lifespan in rats). In addition, they insisted i keeping parts in the discussion that are not directly linked to any of the papers' findings.

---

## [Author Response]

The following is the authors’ response to the previous reviews

**Reviewer #1 (Public Review):**
Summary:Previous studies have shown that treatment with 17α-estradiol (a stereoisomer of the 17β-estradiol) extends lifespan in male mice but not in females. The current study by Li et al, aimed to identify cell-specific clusters and populations in the hypothalamus of aged male rats treated with 17α-estradiol (treated for 6 months). This study identifies genes and pathways affected by 17α-estradiol in the aged hypothalamus.Strengths:Using single-nucleus transcriptomic sequencing (snRNA-seq) on the hypothalamus from aged male rats treated with 17α-estradiol they show that 17α-estradiol significantly attenuated age-related increases in cellular metabolism, stress, and decreased synaptic activity in neurons.

Thanks.

Moreover, sc-analysis identified GnRH as one of the key mediators of 17α-estradiol's effects on energy homeostasis. Furthermore, they show that CRH neurons exhibited a senescent phenotype, suggesting a potential side effect of the 17α-estradiol. These conclusions are supported by supervised clustering by neuropeptides, hormones, and their receptors.

Thanks.

Weaknesses:However, the study has several limitations that reduce the strength of the key claims in the manuscript. In particular:(1) The study focused only on males and did not include comparisons with females. However, previous studies have shown that 17α-estradiol extends lifespan in a sex-specific manner in mice, affecting males but not females. Without the comparison with the female data, it's difficult to assess its relevance to the lifespan.

This study was originally designed based on previous findings indicating that lifespan extension is only effective in males, leading to the exclusion of females from the analysis. The primary focus of our research was on the transcriptional changes and serum endocrine alterations induced by 17α-estradiol in aged males compared to untreated aged males. We believe that even in the absence of female subjects, the significant effects of 17α-estradiol on metabolism in the hypothalamus, synapses, and endocrine system remain evident, particularly regarding the expression levels of GnRH and testosterone. Notably, lower overall metabolism, increased synaptic activity, and elevated levels of GnRH and testosterone are strong indicators of health and well-being in males, supporting the validity of our primary conclusions. However, including female controls would enhance the depth of our findings. If female controls were incorporated, we propose redesigning the sample groups to include aged male control, aged female control, aged female treated, aged male treated, as well as young male control, young male treated, young female control, and young female treated. We regret that we cannot provide this data in the short term. Nevertheless, we believe this reviewer’s creative idea presents a valuable avenue for future research on this topic. In this study, we emphasize the role of 17α-estradiol in overall metabolism, synaptic function, GnRH, and testosterone in aged males and underscore the importance of supervised clustering of neuropeptide-secreting neurons in the hypothalamus.

(2) It is not known whether 17α-estradiol leads to lifespan extension in male rats similar to male mice. Therefore, it is not possible to conclude that the observed effects in the hypothalamus, are linked to the lifespan extension.

Thanks for the reminding. 17α-estradiol was reported to extend lifespan in male rats similar to male mice (PMID: 33289482). We have added the valuable reference to introduction in the new version.

(3) The effect of 17α-estradiol on non-neuronal cells such as microglia and astrocytes is not well-described (Figure 1). Previous studies demonstrated that 17α-estradiol reduces microgliosis and astrogliosis in the hypothalamus of aged male mice. Current data suggest that the proportion of oligo, and microglia were increased by the drug treatment, while the proportions of astrocytes were decreased. These data might suggest possible species differences, differences in the treatment regimen, or differences in drug efficiency. This has to be discussed.

We have reviewed reports describing changes in cell numbers following 17α-estradiol treatment in the brain, using the keywords "17α-estradiol," "17alpha-estradiol," and "microglia" or "astrocyte." Only a limited amount of data was obtained. We found one article indicating that 17α-estradiol treatment in Tg (AβPP(swe)/PS1(ΔE9)) model mice resulted in a decreased microglial cell number compared to the placebo (AβPP(swe)/PS1(ΔE9) mice), but this change was not significant when compared to the non-transgenic control (PMID: 21157032). The transgenic AβPP(swe)/PS1(ΔE9) mouse model may differ from our wild-type aging rat model in this context.

Moreover, the calculation of cell numbers was based on visual observation under a microscope across several brain tissue slices. This traditional method often yields controversial results. For example, oligodendrocytes in the corpus callosum, fornix, and spinal cord have been reported to be 20-40% more numerous in males than in females based on microscopic observations (PMID: 16452667). In contrast, another study found no significant difference in the number of oligodendrocytes between sexes when using immunohistochemistry staining (PMID: 18709647). Such discrepancies arising from traditional observational methods are inevitable.

We believe the data presented in this article are reliable because the cell number and cell ratio data were derived from high-throughput cell counting of the entire hypothalamus using single-cell suspension and droplet wrapping (10x Genomics).

(4) A more detailed analysis of glial cell types within the hypothalamus in response to drugs should be provided.

We provided more enrichment analysis data of differentially expressed genes between Y, O, and O.T in microglia and astrocytes in Figure 2—figure supplement 3. In this supplemental data, we found unlike that in neurons, Micro displayed lower levels of synapse-related cellular processes in O.T. compared to O.

(5) The conclusion that CRH neurons are going into senescence is not clearly supported by the data. A more detailed analysis of the hypothalamus such as histological examination to assess cellular senescence markers in CRH neurons, is needed to support this claim.

We also noted the inappropriate claim and have changed "senescent phenotype" to "stressed phenotype" and "abnormal phenotype" in both the abstract and results sections. The stressed phenotype could be induced by heightened functional activity in the cells, potentially indicating higher cellular activity. The GnRH and CRH neurons discussed in this paper may represent such a case, as illustrated by the observed high serum GnRH, testosterone, and cortisol levels. This revision suggestion is highly valuable and constructive for our understanding of the unique physiological characteristics revealed by these data.

**Reviewer #2 (Public Review):**
Summary:Li et al. investigated the potential anti-ageing role of 17α-Estradiol on the hypothalamus of aged rats. To achieve this, they employed a very sophisticated method for single-cell genomic analysis that allowed them to analyze effects on various groups of neurons and non-neuronal cells. They were able to sub-categorize neurons according to their capacity to produce specific neurotransmitters, receptors, or hormones. They found that 17α-Estradiol treatment led to an improvement in several factors related to metabolism and synaptic transmission by bringing the expression levels of many of the genes of these pathways closer or to the same levels as those of young rats, reversing the ageing effect. Interestingly, among all neuronal groups, the proportion of Oxytocin-expressing neurons seems to be the one most significantly changing after treatment with 17α-Estradiol, suggesting an important role of these neurons in mediating its anti-ageing effects. This was also supported by an increase in circulating levels of oxytocin. It was also found that gene expression of corticotropin-releasing hormone neurons was significantly impacted by 17α-Estradiol even though it was not different between aged and young rats, suggesting that these neurons could be responsible for side effects related to this treatment. This article revealed some potential targets that should be further investigated in future studies regarding the role of 17α-Estradiol treatment in aged males.Strengths:(1) Single-nucleus mRNA sequencing is a very powerful method for gene expression analysis and clustering. The supervised clustering of neurons was very helpful in revealing otherwise invisible differences between neuronal groups and helped identify specific neuronal populations as targets.

Thanks.

(2) There is a variety of functions used that allow the differential analysis of a very complex type of data. This led to a better comparison between the different groups on many levels.

Thanks.

(3) There were some physiological parameters measured such as circulating hormone levels that helped the interpretation of the effects of the changes in hypothalamic gene expression

Thanks.

Weaknesses(1) One main control group is missing from the study, the young males treated with 17α-Estradiol.

Given that the treatment period lasts six months, which extends beyond the young male rats' age range, we aimed to investigate the perturbation of 17α-Estradiol on the normal aging process. Including data from young males could potentially obscure the treatment's effects in aged males due to age effects, though similar effects between young and aged animals may exist. Long-term treatment of hormone may exert more developmental effects on the young than the old. Consequently, we decided to exclude this group from our initial sample design. We apologize for this omission.

(2) Even though the technical approach is a sophisticated one, analyzing the whole rat hypothalamus instead of specific nuclei or subregions makes the study weaker.

The precise targets of 17α-Estradiol within the hypothalamus remain unresolved. Selecting a specific nucleus for study is challenging. The supervised clustering method described in this manuscript allows us to identify the more sensitive neuron subtypes influenced by 17α-Estradiol and aging across the entire hypothalamus, without the need to isolate specific nuclei in a disturbed hypothalamic environment.

(3) Although the authors claim to have several findings, the data fail to support these claims. You may mean the claim as the senescent phenotype in Crh neuron induced by 17a-estradiol.

Thanks. We have changed the "senescent phenotype" to "stressed phenotype" in the abstract and results to avoid such claim. The stressed phenotype may be induced by heightened functional activity in the cells, potentially indicating higher cellular activity.

(4) The study is about improving ageing but no physiological data from the study demonstrated such a claim with the exception of the testes histology which was not properly analyzed and was not even significantly different between the groups.

The primary objective of this study is to elucidate the effects of 17α-Estradiol on the endocrine system in the aging hypothalamus; exploring anti-aging effects is not the main focus. From the characteristics of the aging hypothalamus, we know that down-regulated GnRH and testosterone levels, along with elevated mTOR signaling, are indicators of aging in these organs from previous publications (PMID: 37886966, PMID: 37048056, PMID: 22884327). The contrasting signaling networks related to metabolism and synaptic processes significantly differentiate young and aging hypothalami, and 17α-Estradiol helps rebalance these networks, suggesting its potential anti-aging effects.

(5) Overall, the study remains descriptive with no physiological data to demonstrate that any of the effects on hypothalamic gene expression are related to metabolic, synaptic, or other functions.

The study focuses on investigating cellular responses and endocrine changes in the aging hypothalamus induced by 17α-estradiol, utilizing single-nucleus RNA sequencing (snRNA-seq) and a novel data mining methodology to analyze various neuron subtypes. It is important to note that this study does not mainly aim to explore the anti-aging effects. Consequently, we have revised the claim in the abstract from “the effects of 17α-estradiol in anti-aging in neurons” to “the effects of 17α-estradiol on aging neurons.” We observed that the lower overall metabolism and increased expression levels of cellular processes in the synapses align with findings previously reported regarding 17α-estradiol. To address the lack of physiological data and the challenges in measuring multiple endocrine factors due to their volatile nature, we employed several bidirectional Mendelian analyses of various genome-wide association study (GWAS) data related to these serum endocrine factors to identify their mutual causal effects.

**Reviewing Editor Comment:**
Based on the Public Reviews and Recommendations for Authors, the Reviewers strongly recommend that revisions include an experimental demonstration of the physiological effects of the treatment on ageing in rats as well as the CRH-senescence link. Additional analysis of the glia would greatly strengthen the study, as would inclusion of females and young male controls. The important point was also raised that the work linking 17a-estradiol was performed in mice, and the link with lifespan in rats is not known. Discussion of this point is recommended.

We thank the reviewers for their constructive feedback. Regarding the recommendations in the Public Reviews and Recommendations for Authors:

a) Physiological effects & CRH-senescence link:

We acknowledge that 17α-estradiol has been reported to extend lifespan in male rats, consistent with findings in male mice (PMID: 33289482). This point has now been noted in the Introduction. We regret that further experimental validation of the treatment's physiological effects on aging in rats was beyond the scope of this study.

b) Phenotype terminology:

In response to concerns about the "senescent" characterization of CRH neurons, we have revised this terminology to "stressed phenotype" throughout the abstract and results. While we were unable to conduct additional experiments to confirm senescence markers, this revised description better reflects the heightened cellular activity observed (as evidenced by elevated serum GnRH and testosterone levels), without implying confirmed senescence.

c) Glial cell analysis:

To address questions about glial cell function during treatment, we have added new enrichment analysis data of differentially expressed genes in microglia and astrocytes from young (Y), old (O), and old treated (O.T) groups in Figure 2—figure supplement 3. This analysis reveals that microglia exhibit contrasting synaptic-related cellular processes compared to total neurons.

d) Female and young controls:

We sincerely apologize for the absence of female subjects and young male controls in the current study. The reviewers' suggestion to examine the male-specific effects of 17α-estradiol using female controls represents an excellent direction for future research, which we plan to pursue in upcoming studies.

**Reviewer #2 (Recommendations For The Authors):**
General comments:(1) The manuscript is very hard to read. Proofreading and editing by software or a professional seems necessary. The words "enhanced", "extensive" etc. are not always used in the right way.

Thanks for the suggestion. We have revised the proofreading and editing. The words "enhanced" and "extensive" were also revised in most sentences.

(2) The numbers of animals and samples are not well explained. Is it 9 rats overall or per group? If there are 8 testes samples per group, should we assume that there were 4 rats per group? The pooling of the hypothalamic how was it done? Were all the hypothalamic from each group pooled together? A small table with the animals per group and the samples would help.

We appreciate your reminder regarding the initial mistake in our manuscript preparation. In the preliminary submission, we reported 9 rats based solely on sequencing data and data mining. The revised version (v1) now includes additional experimental data, with an effective total of 12 animals (4 per group). Unfortunately, we overlooked updating this information in the v1 submission. We have since added detailed information in the Materials and Methods sections: Animals, Treatment and Tissues, and snRNA-seq Data Processing, Batch Effect Correction, and Cell Subset Annotation.

(3) The Clustering is wrong. There are genes in there that do not fall into any of the 3 categories: Neurotransmitters, Receptors, Hormones.

We acknowledge the error in gene clustering and have implemented the following corrections:

(a) The description has been updated to state: 'Vast majority of these subtypes were clustered by neuropeptides, hormones, and their receptors among all neurons.'

(b) Genes not belonging to these three categories have been substantially removed.

(c) The neuropeptide category (now including several growth hormones) has been expanded to 104 genes, while their corresponding receptors (including several sex hormone receptors) now comprise 105 genes.

(4) The coloring of groups in the graphs is inconsistent. It must be more homogeneous to make it easier to identify.

We have changed the colors of groups in Fig. 1D to make the color of cell clusters consistent in Fig. 1A-D.

(5) The groups c1-c4 are not well explained. How did the authors come up with these?

We have added more descriptions of c1-c4 in materials and methods in the new version.

(6) In most cases it's not clear if the authors are talking about cell numbers that express a certain mRNA, the level of expression of a certain mRNA, or both. They need to do a better job using more precise descriptions instead of using general terms such as "signatures", "expression profiles", "affected neurons" etc. It is very hard to understand if the number of neurons is compared between the groups or the gene expression.

We have changed the "signatures" to "gene signatures" to make it more accurate in meaning. The "affected neurons" were also changed to "sensitive neurons". But sorry that we were not able to find better alternatives to the "expression profiles".

(7) Sometimes there are claims made without justification or a reference. For example, the claim about the senescence of CRH neurons due to the upregulation of mitochondrial genes and downregulation of adherence junction genes (lines 326-328) should be supported by a reference or own findings.

The "senescence" here is not appropriate. We have changed it to "stressed phenotype" or "aberrant changes" in abstract and results.

(8) Young males treated with Estradiol as a control group is necessary and it is missing.

Your suggestion is appreciated; however, the treatment duration for aged mice (O.T) was set at 6 months, while the young mice were only 4 months old. This disparity makes it challenging to align treatment timelines for the young animals. The primary aim of this study is to investigate the perturbation of 17α-estradiol on the aging process, and any distinct effects due to age effect observed in young males might complicate our understanding of its role in aged males, though similar endocrine effects may exist in the young animals. Long-term treatment of hormone may exert more developmental effects on the young than the old. Therefore, we made the decision to exclude the young samples in our initial study design. We apologize for any confusion this may have caused.

Specific Comments:

Line 28: "elevated stresses and decreased synaptic activity": Please make this clearer. Can't claim changes in synaptic activity by gene expression.

We have changed it to "the expression level of pathways involved in synapse"

Line 32: "increased Oxytocin": serum Oxytocin.

We have added the “serum”.

Line 52 - 54: Any studies from rats?

Thanks. In rats there is also reported that 17α-estradiol has similar metabolic roles as that in mice (PMID: 33289482) and we have added it to the refences. It’s very useful for this manuscript.

Line 62 - 65: It wasn't investigated thoroughly in this paper so why was it suggested in the introduction?

We have deleted this sentence as being suggested.

Line 70: "synaptic activity" Same as line 28.

We have changed it to "pathways involved in synaptic activity".

Line 79: Why were aged rats caged alone and young by two? Could that introduce hypothalamic gene expression effects?

The young males were bred together in peace. But the aged males will fight and should be kept alone.

Lines 78, 99, 109-110: It is not clear how many animals per group were used and how many samples per group were used separately and/or grouped. Please be more specific.

We have added these information to Materials and methods/Animals, treatment and tissues and Materials and methods/snRNA-seq data processing, batch effect correction, and cell subset annotation.

Line 205: "in O" please add "versus young.".

We have changed accordingly.

Line 207: replace "were" with "was"

We have alternatively changed the "proportion" to "proportions".

Line 208: replace "that" with "compared to" and after "in O.T." add "compared to?"

We have changed accordingly.

Line 223: "O.T." compared to what? Figure?

We have changed it accordingly.

Line 227: Figure?

We have added (Figure 1E) accordingly.

Line 229: "synaptic activity" Same as line 28.

We have revised it.

Line 235: "synaptic activity" and "neuropeptide secretion" Same as line 28.

We have revised it.

Line 256:" interfered" please revise.

We changed to "exerted".

Line 263: "on the contrary" please revise.

We have changed "on the contrary" to "opposite".

Line 270: "conversed" did you mean "conserved"?

We have changed "conversed" to "inversed".

Line 296-298: Please explain. Why would these be side effects?

It’s hard to explain, therefore, we deleted the words "side effects".

Line 308: "synaptic activity" Same as line 28.

We have changed it to "expression levels of synapse-related cellular processes".

Line 314: "and sex hormone secretion and signaling"Isn't this expected?

Yes, it is expected. We have added it to the sentence "and, as expected, sex hormone secretion and signaling".

Line 325-328: Why is this senescence? Reference?

We have added “potent” to it.

Line 360-361: This doesn't show elevated synaptic activity.

"elevated synaptic activity" was changed to "The elevated expression of synapse-related pathways"

Line 363-364: "Unfortunately" is not a scientific expression and show bias.

We have changed it to "Notably".

Line 376: Similar as above.

Yes, we have change it to "in contrast".

Lines 382-385: This is speculation. Please move to discussion.

Sorry for that. We think the causal effects derived from MR result is evidence. As such, we have not changed it.

Line 389: Please revise "hormone expressing".

We have changed it accordingly.

Line 401: Isn't this effect expected due to feedback inhibition of the biochemical pathway? Please comment.

The binding capability of 17alpha-estradiol to estrogen receptors and its role in transcriptional activation remain core questions surrounded by controversy. Earlier studies suggest that 17alpha-estradiol exhibits at least 200 times less activity than 17beta-estradiol (PMID: 2249627, PMID: 16024755). However, recent data indicate that 17alpha-estradiol shows comparable genomic binding and transcriptional activation through estrogen receptor α (Esr1) to that of 17beta-estradiol (PMID: 33289482). Additionally, there is evidence that 17alpha-estradiol has anti-estrogenic effects in rats (PMID: 16042770). These findings imply possible feedback inhibition via estrogen receptors. Furthermore, 17alpha-estradiol likely differs from 17beta-estradiol due to its unique metabolic consequences and its potential to slow aging in males, an effect not attributed to 17beta-estradiol. For instance, neurons are also targets of 17alpha-estradiol, with Esr1 not being the sole target (PMID: 38776045). Intriguingly, neurons expressing Ar and Esr1 ranked among the top 20 most perturbed receptor subtypes during aging (O vs Y), but were no longer ranked in this group following treatment (O.T vs Y and O.T vs O comparisons). This indicates that 17α-estradiol administration attenuated age-associated perturbation in these neuronal subtypes, which may be a consequence of potential feedback (Figure 3D). Nevertheless, the precise effective targets of 17alpha-estradiol are still unresolved.

Line 409: This conclusion cannot be made because the effect is not statistically significant. Can say "trend" etc.

Thanks for the recommendation. We have added "potential" in front of the conclusion.

Line 426: "suggesting" please revise.

sorry, it’s a verb.

Lines 426-428: This is speculation. Please move to discussion.

The elevated GnRH levels in O.T., observed through EIA analysis, suggest a deduction regarding the direct causal effects of 17alpha-estradiol on various endocrine factors related to feeding, energy homeostasis, reproduction, osmotic regulation, stress response, and neuronal plasticity through MR analysis. Thus, we have not amended our position. We apologize for any confusion.

Lines 431-432: improved compared to what?

The statement have been revised as " The most striking role of 17α-estradiol treatment revealed in this study showed that HPG axis was substantially improved in the levels of serum Gnrh and testosterone".

Line 435: " Estrogen Receptor Antagonists". Please revise.

Thanks for the recommendation. We have changed it to "estrogen receptor antagonists".

Line 438" "Secrete". Please revise

Sorry, it is "secret".

Lines 439-449: None of this has been demonstrated. Please remove these conclusions.

We appreciate the reviewer's scrutiny regarding lines 439-449. While these statements should not be interpreted as definitive conclusions from our current data, we propose they serve as clinically relevant discussion points worthy of exploration. Our findings demonstrate 17α-estradiol's role in modulating testosterone levels in aged males. This mechanistic insight warrants consideration of its therapeutic potential for age-related hypogonadism - a hypothesis we believe merits discussion given the compound's specific endocrine effects.

Lines 450-457: No females were included in this study. Why? Also, why is this discussed? It is relevant but doesn't belong in this manuscript since it was not studied here.

Testosterone levels are crucial for male health, while estradiol levels are essential for the health and fertility of females. Previous studies have demonstrated that 17α-estradiol does not contribute to lifespan extension in females. Given the effects of 17α-estradiol on males—specifically, its role in promoting testosterone and reducing estradiol levels—we believe it is important to discuss the potential sex-biased effects of 17α-estradiol, as this could inform future investigations. We have refined this section to clarify that these points represent mechanistic hypotheses derived from our male data and existing literature, not conclusions about unstudied female physiology. This framing maintains the discussion's scientific value while respecting the study's scope.

Lines 458-459: This was not demonstrated in this article. Please remove.

We have restricted the claim to "expression level of energy metabolism in hypothalamic neurons".

Line 464: "Promoted lifespan extension" Not demonstrated. Please remove.

At the end of the sentence it was revised as "which may be a contributing factor in promoting lifespan extension".

Line 466: "Showed" No.

The whole sentence was deleted in the new version.

Line 483: "the sex-based effects". Not studied here.

Since the changes in testosterone levels are significant in this dataset and this hormone has a sex-biased nature, we find it worthwhile to suggest this as a topic for future investigation. We have added "which needs further verification in the future" at the end of this sentence.